# A single-cell transcriptome atlas of pig skin characterizes anatomical positional heterogeneity

Qin Zou[1†], Rong Yuan[2†], Yu Zhang[3†], Yifei Wang[1], Ting Zheng[1], Rui Shi[1], Mei Zhang[1], Yujing Li[1], Kaixin Fei[1], Ran Feng[1], Binyun Pan[1], Xinyue Zhang[1], Zhengyin Gong[1], Li Zhu[4], Guoqing Tang[4], Mingzhou Li[4], Xuewei Li[4], Yanzhi Jiang[1*]

[1]Department of Zoology, College of Life Science, Sichuan Agricultural University, Ya'an, China; [2]Chengdu Livestock and Poultry Genetic Resources Protection Center, Chengdu, China; [3]BGI Beijing Genome Institute, Beijing, China; [4]College of Animal Science and Technology, Sichuan Agricultural University, Chengdu, China

*For correspondence:
jiangyz04@163.com

[†]These authors contributed equally to this work

Competing interest: The authors declare that no competing interests exist.

**Abstract** Different anatomical locations of the body skin show differences in their gene expression patterns depending on different origins, and the inherent heterogeneous information can be maintained in adults. However, highly resolvable cellular specialization is less well characterized in different anatomical regions of the skin. Pig is regarded as an excellent model animal for human skin research in view of its similar physiology to human. In this study, single-cell RNA sequencing was performed on pig skin tissues from six different anatomical regions of Chenghua (CH) pigs, with a superior skin thickness trait, and the back site of large white (LW) pigs. We obtained 233,715 cells, representing seven cell types, among which we primarily characterized the heterogeneity of the top three cell types, including smooth muscle cells (SMCs), endothelial cells (ECs), and fibroblasts (FBs). Then, we further identified several subtypes of SMCs, ECs, and FBs, and discovered the expression patterns of site-specific genes involved in some important pathways such as the immune response and extracellular matrix (ECM) synthesis in different anatomical regions. By comparing differentially expressed genes of skin FBs among different anatomical regions, we considered TNN, COL11A1, and INHBA as candidate genes for facilitating ECM accumulation. These findings of heterogeneity in the main skin cell types from different anatomical sites will contribute to a better understanding of inherent skin information and place the potential focus on skin generation, transmission, and transplantation, paving the foundation for human skin priming.

## Editor's evaluation

This valuable manuscript provides a single-cell RNA sequencing analysis of adult pig skin from different species and anatomical regions. The evidence supporting the conclusions is compelling, with identification of molecular and cellular differences in pig skin, including analysis of regional and species-specific gene signatures.

## Introduction

The issue of how inherent information contributes to anatomical site-specific differences has inspired extensive exploration, and the pattern formation of spatial arrangement depends on the expression control of specific genes with a cell type (*Rinn et al., 2006*). The cellular specialization of anatomical site-specific pattern is determined in the embryo, and the inherent information could also be maintained throughout adulthood along with continual self-renewal tissues (*Rinn et al., 2006*). Some

cellular special information of anatomical site-specific patterns in other tissues has been discovered, such as in the heart (*Litviňuková et al., 2020*) and muscle (*De Micheli et al., 2020*), but the inherent information of cellular specialization is less well understood in physiologically different anatomical skin regions.

Skin is the largest organ, providing a physical, chemical, and biological barrier for the body. It consists of the upper epidermis and the lower dermis layers separated by the basement membrane, with unambiguous spatial patterns of morphological and functional specialization (*Simpson et al., 2011*). Embryological studies have shown that anatomical positional-specific pattern is provided by the stroma, which is composed of extracellular matrix (ECM) and mesenchymal or dermal cells during embryogenesis (*Rinn et al., 2006*). Pioneering studies discovered that the different anatomical regions of the body skin dermis arose from different origins. The dorsum dermis originates from the dermato-myotome, while the ventral and face dermis derive from the lateral plate mesoderm and the neural crest, respectively (*Jinno et al., 2010*; *Ohtola et al., 2008*; *Wong et al., 2006*). In adults, dermal cells confer the positional identity and memory for skin patterning and function (*Driskell and Watt, 2015*), raising the issue of what regional discrepancy could be maintained against plentiful cellular turnover in skin.

The dermis is mainly composed of resident fibroblasts (FBs), smooth muscle cells (SMCs), endothelial cells (ECs), and immune cells, and these skin cells provide structure, strength, flexibility, and defense to the skin (*Driskell and Watt, 2015*). FBs, the main cell type in the dermis, are responsible for the collagen deposits and elastic fiber formation of the ECM (*Parsonage et al., 2005*), participating in skin morphogenesis, homeostasis, and various physiological and pathological mechanisms, including skin development, aging, healing, and fibrosis (*Auxenfans et al., 2009*; *Driskell et al., 2013*; *Driskell and Watt, 2015*). SMCs, which form blood vessels and arrector pili muscle (APM), play a critical role in controlling blood distribution as well as maintaining the structural integrity of the blood vessels and APM in skin (*Driskell et al., 2013*; *Liu and Gomez, 2019*). ECs organize the vascular plexus, which plays a predominant role in vascular remodeling, metabolism, and the immune response in the dermis, and EC metabolism is tightly connected to barrier integrity, immune and cellular crosstalk with SMCs (*Cantelmo et al., 2016*; *Miyagawa et al., 2019*; *Tombor et al., 2021*). Moreover, during formation and development of the skin, the dermal reaction is realized by cell–cell communication, and dynamic interactions between cells and ECM, as well as regulatory factors (*Driskell and Watt, 2015*).

The pig is used as a model animal to research human skin biology because of its similar pathological and physiological skin attributes to those of human skin (*Khiao In et al., 2019*). The Chenghua (CH) pig, a novel Chinese indigenous population with superior skin thickness and strong FBs activity (*Zou et al., 2022*; *Zou et al., 2023*), is considered as a potential model animal for researching mammalian skin biology. Here, to reveal the anatomical positional heterogeneity of the skin, single-cell RNA sequencing was performed on pig skin tissues from six anatomical regions of CH pigs and the back site of large white (LW) pigs. We obtained a well-resolved single-cell transcriptome atlas of 233,715 cells and identified seven cell types with unique gene expression signatures. In our datasets, we focused on the top three cell types, including SMCs, ECs, and FBs. SMCs revealed the signature of contractile SMCs, mesenchymal-like phenotype, and macrophage-like phenotype and presented the expression patterns of site-specific genes related to ECM-integrins and immune response in different skin anatomical sites. ECs were classified into four EC phenotypes, and the gene expression patterns, which are related to integrins, immunity, and metabolism, were explored across different skin anatomical sites. Moreover, based on these comparative differentially expressed genes (DEGs) of FBs, we identified three subtypes among different regions and found that TNN, COL11A1, and INHBA might be candidate genes for ECM accumulation. Taken together, the present data offer a comprehensive understanding of the single-cell atlas that displays the cellular inherent information of anatomical site-specific patterns in skin, supporting future exploration as a baseline for healthy and morbid human skin.

# Results

## Single-cell transcriptome profiling identified different skin anatomical sites in CH pig

To characterize an overview of single-cell transcriptomic atlas of pig skin tissues from different anatomical sites, we sampled CH pig skin tissues on the head, ear, shoulder, back, abdomen, and leg from three female 180-day-old individuals, then applied scRNA-seq of 18 isolated skin cell samples (*Figure 1A*). After stringent cell filtration, we obtained a total of 215,274 cells, which were globally visualized with 21 cell clusters in the t-SNE plot (*Figure 1B*). On average, 956 genes and 2687 unique molecular identifiers (UMIs) per cell were detected (*Figure 1—figure supplement 1A and B*). Twenty-one cell clusters were identified according to the expression matrix of marker genes for each cluster and were shown in the heatmap basing on the top 12 marker genes for each cluster (*Figure 1C*). The 21 cell clusters constituted seven cell types, of which the SMCs (clusters 0, 2, 5, 6, and 13) were marked by MYH11 and ACTA2, ECs (clusters 3, 4, 7,10, and 11) were marked by PECAM1 and APOA1, FBs (clusters 1, 8, 9, and 12) were expressed by LUM and POSTN, myeloid dendritic cells (MDCs) (clusters 14, 16, and 18) were labeled by BCL2A1 and CXCL8, T cells (TCs) (cluster 15) were highly expressed by RHOH and SAMSN1, keratinocytes (KEs) (cluster 17) were tabbed by KRT5 and S100A2, and epidermal stem cells (ESCs) (clusters 19 and 20) were stamped by TOP2A and EGFL8 (*Figure 1D and E* and *Figure 1—figure supplement 1C*).

The distribution ratio of these cell types was visualized among total skin data consisting of 42.9% SMCs, 28.1% ECs, 24.6% FBs, 2.5% MDCs, 0.9% TCs, 0.6% KEs, and 0.3% ESCs, which were similar to the distribution ratio for the main cell types in different skin regions (*Figure 1F*). In addition, the cell number and types among the six anatomical skin sites were comparable, which indicated that the cell types displayed subtle differences, but cell number per cell type varied significantly (*Figure 1—figure supplement 2*). The marker genes for each cell type revealed the dominant transcriptional features and enriched pathways relevant to their distinct physical functions. Significant examples of Gene Ontology (GO) function terms were involved in ECM structural constituent or collagen binding for FBs, actin binding or structural constituent of muscle for SMCs, and ECM structural constituent or collagen binding for ECs (*Figure 1G*). Meanwhile, Kyoto Encyclopedia of Genes and Genomes (KEGG) pathways were prominently attributed to FBs such as protein digestion and absorption or ECM–receptor interaction, ECs involved in cell adhesion molecules or the Rap1 signaling pathway, and SMCs including the NF-kappa B signaling pathway or the TNF signaling pathway (*Figure 1G*).

Moreover, given the potential cross-species comparisons, we implemented overlapping skin cell atlases among pig, human, and mouse using a t-SNE plot (*Figure 1—figure supplement 3A*). The captured gene and UMI counts were more advantageous for human skin cells (*Figure 1—figure supplement 3B*). The cell types were similar for the three species, while the percentage of cell types was different such as SMCs, ECs, or KEs (*Figure 1—figure supplement 3A and C*). Some marker genes of skin tissue were shown on the heatmap and dot plots, which examined the shared or species-specific genes in all cell types among the three species (*Figure 1—figure supplement 3D and E*). When discounting the unique skin thickness of CH pig resulting in this ratio discrepancy of the cell types such as excessive cell number of SMCs and ECs, dominantly originated from the vessel bed, we suggested that the pig skin tissue could be considered as the human skin model at single-cell levels for research purposes.

## Heterogeneity of skin SMCs in different anatomical sites

SMCs play critical roles in forming blood vessels and APM in skin tissues (*Driskell et al., 2013*; *Liu and Gomez, 2019*); however, no study has uncharacterized skin SMCs at single-cell resolution. Here, we interrogated the heterogeneity and function of cutaneous SMCs. The t-SNE analysis divided SMCs into five subpopulations (clusters 0, 2, 5, 6, and 13) (*Figure 2A*), in which the MYH11 and ACTA2 marker genes were used for the immunohistochemistry staining of skin sections to validate the SMCs' microanatomical sight (*Figure 2B*). Meanwhile, GO functional analysis was performed on the highly expressed genes for each cluster (*Figure 2C*). Clusters 0 and 13 predominantly took part in structural constituent of muscle, acting filament binding, and acting binding. The engagement of main inflammatory response and chemokine activity belonged to clusters 2, 5, and 6, of which cluster 2 was also involved in collagen binding and metallopeptidase activity. These results implied that SMCs

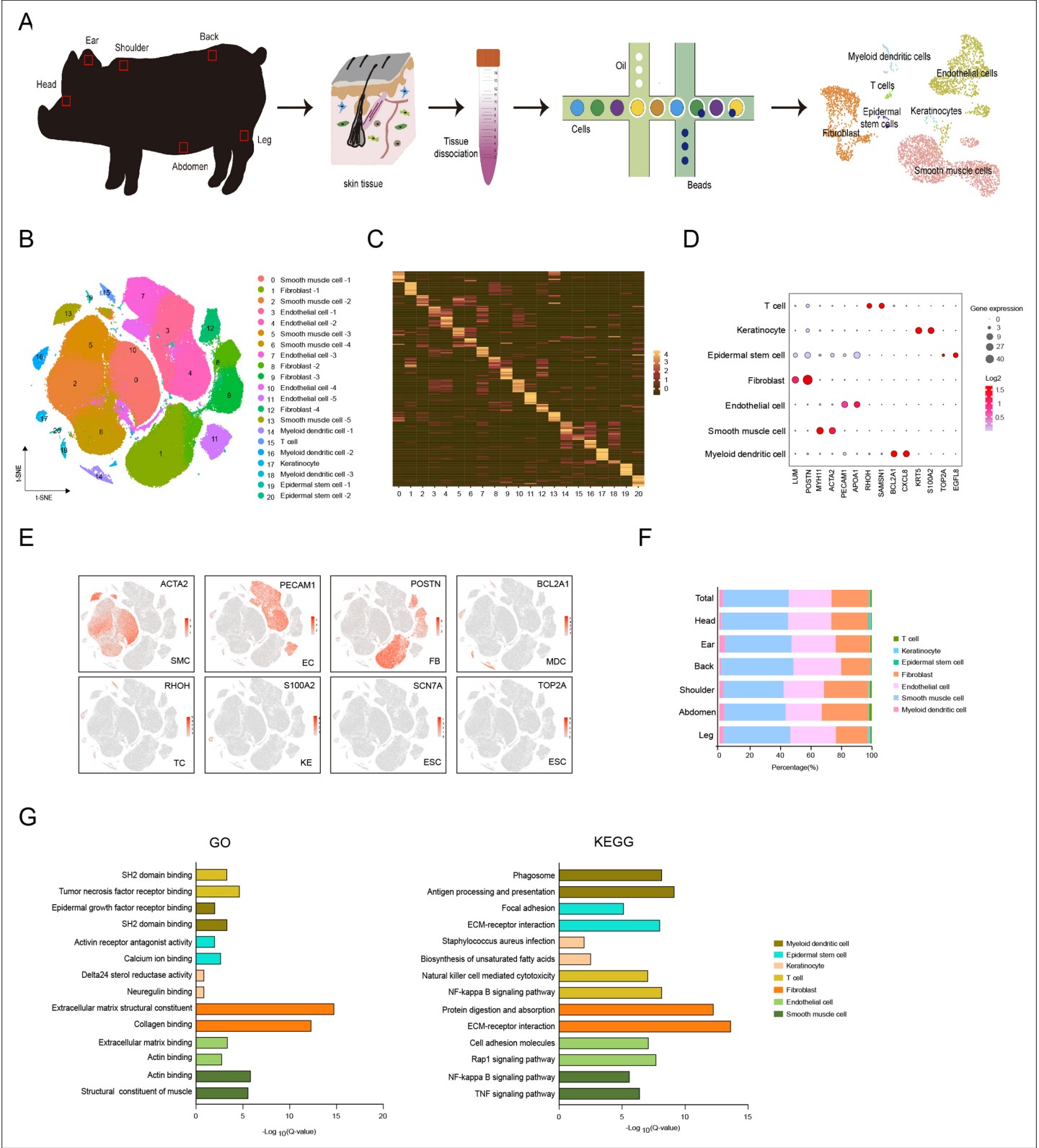

**Figure 1.** Single-cell transcriptome profiling of different skin anatomical sites in CH pig. (**A**) Flowchart overview of skin single-cell RNA sequencing from different anatomical skin regions of CH pigs. (**B**) The t-SNE plot visualization showing 21 clusters of annotated cell types from CH pig skin. (**C**) Heatmap showing the top 12 highly expressed genes from each cluster. Each column represents a cluster, each row represents a gene. Light yellow shows the maximum expression level of genes, and deep green shows no expression. (**D**) Dot plot showing the two representative genes for each cell type. Color

*Figure 1 continued on next page*

*Figure 1 continued*

indicates the log$_2$ value, and circle size indicates gene expression level. (**E**) The marker genes for each cell type are distributed on the t-SNE plot. Color indicates gene expression. (**F**) The distribution ratio of cell types for total cells and six different anatomical skin areas. (**G**) The most enriched Gene Ontology (GO) terms and Kyoto Encyclopedia of Genes and Genomes (KEGG) pathways for each cell type. CH, Chenghua; SMC, smooth muscle cell; EC, endothelial cell; FB, fibroblast; MDC, myeloid dendritic cell; TC, T cell; KE, keratinocyte; and ESC, epidermal stem cell.

The online version of this article includes the following source data and figure supplement(s) for figure 1:

**Source data 1.** Source data of marker genes for each cluster in *Figure 1C*.

**Figure supplement 1.** The count of genes/unique molecular identifier (UMI) and the expression of marker genes.

**Figure supplement 1—source data 1.** Source data of the gene/unique molecular identifier (UMI) counts in *Figure 1—figure supplement 1A and B*.

**Figure supplement 2.** The cell types of different skin regions.

**Figure supplement 3.** Comparison of skin cells among human, pig, and mouse species.

play important roles in blood vessel homeostasis and function, partial collagen binding, and immune responses in skin tissue.

Then, we differentiated skin SMCs into other-like cell types. This evidence, combined with GO function analysis and the expression levels of conventional marker genes, such as MYH11 and ACTA2 for SMCs, GUCY1A2, CCL19, FGF7, and ASPN for mesenchymal cells (MECs), and LPL, CCL2, IL6, and CXCL2 for macrophages (MACs), presumed that cluster 2 might be mesenchymal-like phenotype while clusters 5 and 6 might represent macrophage phenotype. To further validate the topography of SMCs phenotypes, the analysis of pseudotime trajectory was performed by Monocle algorithm (*Figure 2D*). The trajectory demonstrated that SMCs experienced a dynamic transition from SMCs to mesenchymal-like phenotype and mesenchymal-like phenotype to macrophage-like phenotype. The sequential dynamics of gene expression with all branches were visualized and showed five gene sets along with expression patterns, which primarily deciphered three cell states (*Figure 2E*). These results discovered that gene sets 1 and 3 showed high expression levels of CTGF, LGR4, FABP4, CCL2, CCL19, and FGF7, and enriched GO terms associated with negative regulation of cell death, intestinal stem cell homeostasis, long-chain fatty acid transport, and immune response, which conformed well to the mesenchymal-like cells. Meanwhile, gene sets 2 and 5 showed high expression levels of MYH11, MYOM1, TPM1, TPM2, SQLE, BTG2, ADIRF, and TGFB3, and enriched GO terms involved in muscle contraction, actin filament organization, and 'de novo' action filament nucleation, which was greatly similar to contractile SMCs. In addition, gene set 4 showed high expression levels of CXCL10, CXCL2, ICAM1, LPL, and IL6, which were mainly gathered in GO terms of cellular response to lipopolysaccharide, cell chemotaxis, and defense response that may represent macrophage-like cells. Additionally, the expression levels of some cell type-specific marker genes in five SMCs clusters were presented (*Figure 2F*). The results showed the high expression levels of MECs-specific genes (GUCY1A2, FGF7, and CCL19) in cluster 2, MACs-specific LPL gene in clusters 5 and 6, and SMCs-specific MYH11 gene in clusters 0 and 13. These results proved our hypothesis that cluster 2 was mesenchymal-like phenotype while clusters 5 and 6 were macrophage-like phenotype.

The cell number of skin SMCs showed a significant difference in different anatomical sites, while the distribution ratio of SMC subpopulations displayed a similar trend (*Figure 3A*). To decode the transcriptomic changes of skin SMCs in different anatomical sites, the DEGs were presented among 15 compared groups by pairwise comparison method (*Figure 3B*). GO analysis showed that the significant enriched terms for upregulated genes between compared groups primarily referred to extracellular region, collagen-containing ECM, and long-chain fatty acid transport, while these downregulated genes between compared groups took part in cytokine activity, CXCR chemokine receptor binding, and positive regulation of T cell migration (*Figure 3C*). The majority of upregulated genes subsisted in back skin compared to other locations, so we implemented KEGG analysis, which was involved in PI3K-Akt signaling pathway, MAPK signaling pathway, immune response, and integration (*Figure 3D*). Then, we chose some genes of related ECM-integrins and immune response to present their expression levels in different skin anatomical sites, which showed that immune response correlated closely with shoulder skin region or ECM-integrins tightly linked to skin locations on the head, back, and shoulder (*Figure 3E*). Moreover, to gain insights into gene expression regulation, we investigated the key transcription factors (TFs), which regulated the DEGs among various compared groups, using single-cell regulatory network inference and clustering (SCENIC). The SCENIC algorithm demonstrated

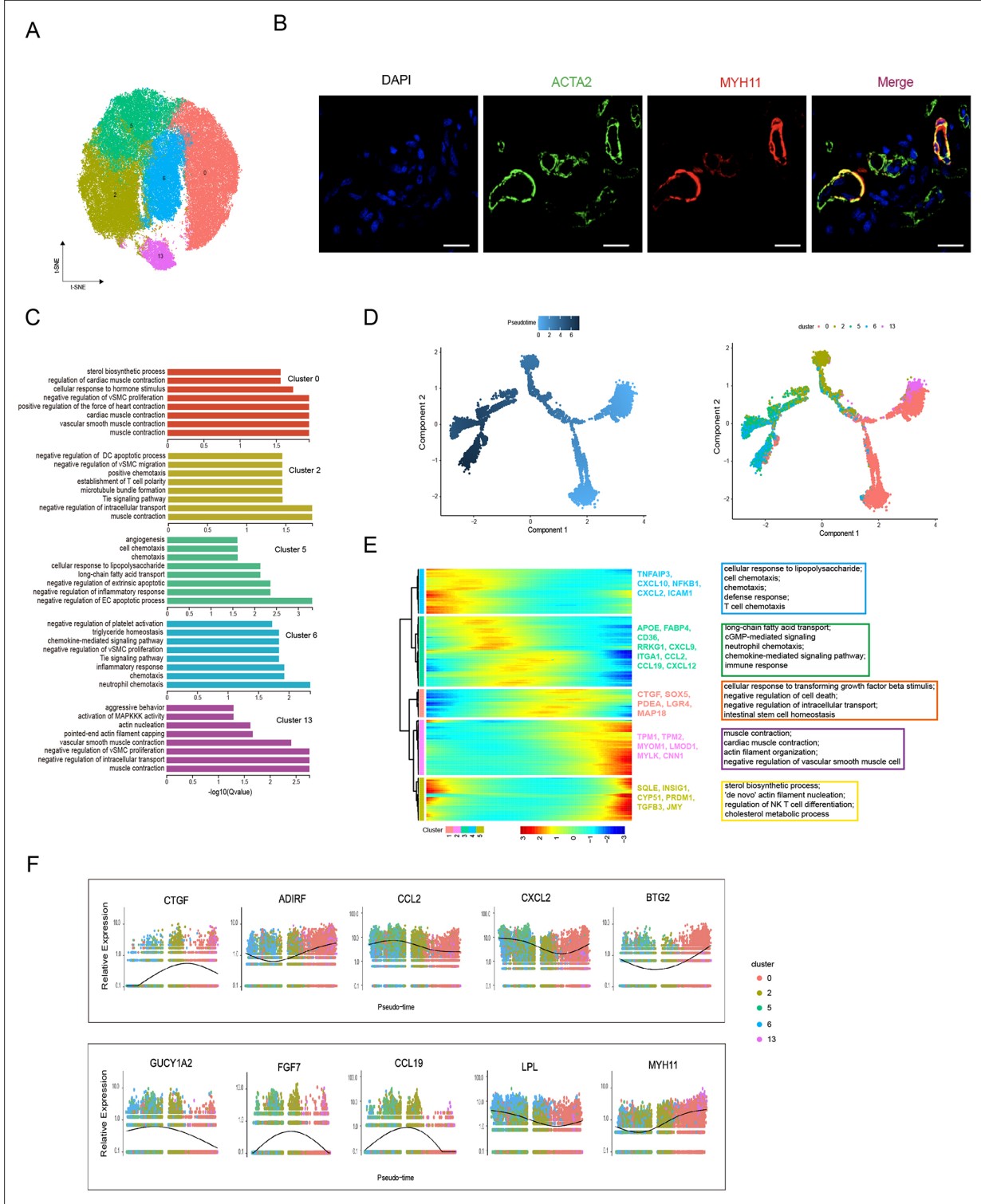

**Figure 2.** Heterogeneity of skin smooth muscle cells (SMCs) in different anatomical sites. (**A**) The t-SNE plot visualization of SMCs including clusters 0, 2, 5, 6, and 13. (**B**) Confocal images showing immunofluorescence staining of ACTA2 (green) and MYH11 (red) in back skin sections, representative markers of SMCs. Scale bar = 50 μm. n = 3. (**C**) The enriched Gene Ontology (GO) terms of biological process for each SMC subpopulation were sorted by q-value. (**D**) Pseudotime ordering of SMC subpopulations using Monocle 2. (**E**) Heatmap illustrating the dynamics of representative differentially expressed genes among SMCs phenotypes, in which the important GO terms relating to biological processes were described. (**F**) These genes' expression along pseudotime in SMC subpopulations.

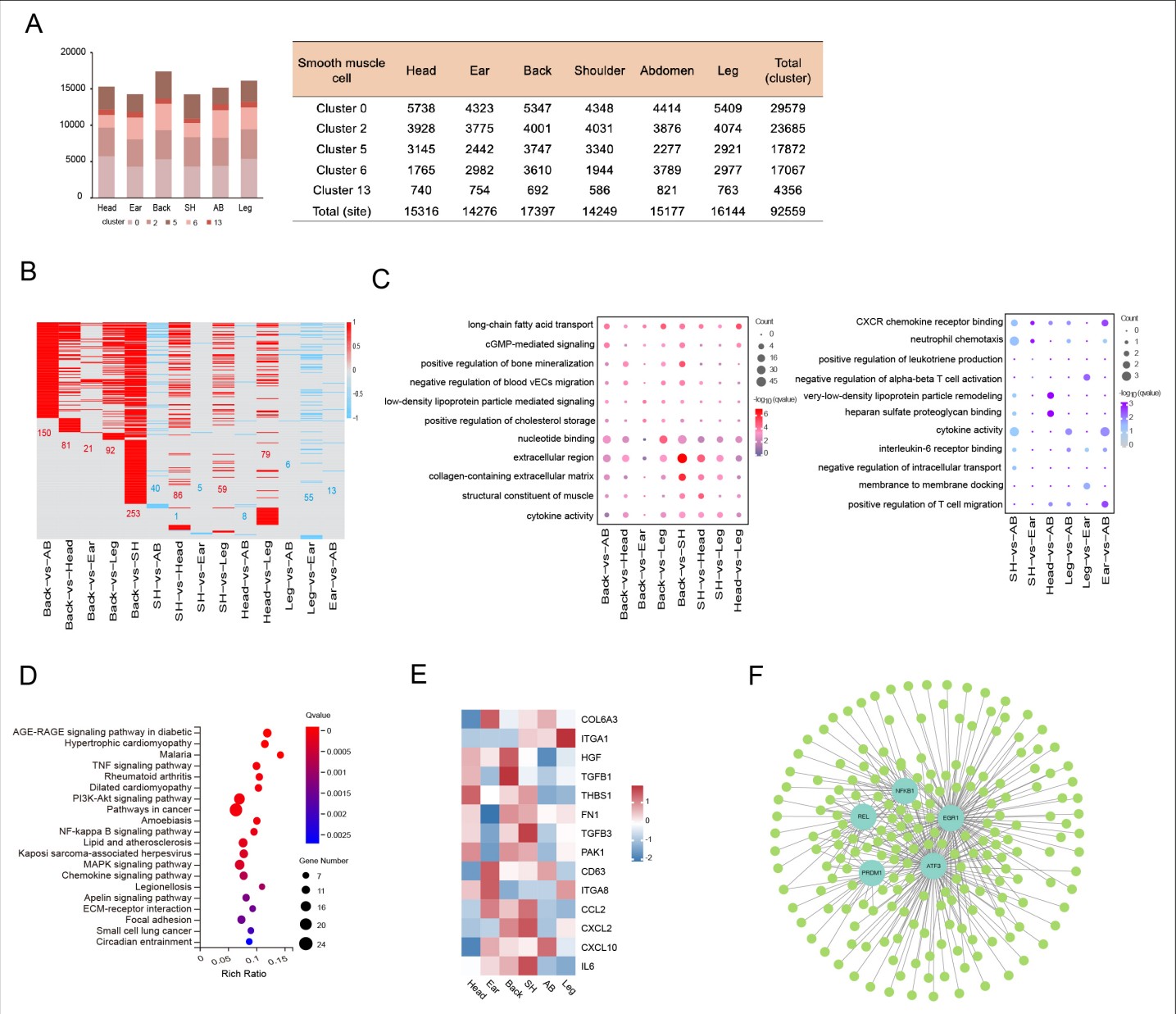

**Figure 3.** Heterogeneity of skin smooth muscle cells (SMCs) in different anatomical sites. (**A**) The cell number of SMC subpopulations in different skin regions. (**B**) Heatmap showing the differentially expressed genes of SMCs in multiple compared groups. Red represents upregulated genes, blue represents downregulated genes, and the number of differentially expressed genes is indicated. (**C**) The enriched Gene Ontology (GO) terms of multiple compared groups. Color indicates q-value, and circle indicates gene counts. (**D**) Kyoto Encyclopedia of Genes and Genomes (KEGG) analysis for upregulated genes of back skin compared to other locations. (**E**) The expression level of genes involved in extracellular matrix (ECM)-integrins and immune response pathways in different skin regions. Red represents high expression of genes. (**F**) Transcriptional regulatory network of differentially expressed genes for SMCs in multiple compared groups. Blue nodes represent regulators and green nodes represent the target genes of regulators.

The online version of this article includes the following source data for figure 3:

**Source data 1.** Source data of the differentially expressed genes of smooth muscle cells (SMCs) in multiple compared groups in *Figure 3B*.

a series of main regulons such as EGR1, ATF3, NFKB1, PRDM1, and REL, and their related target genes (*Figure 3F*). TFs, especially ATF3 and EGR1, primarily regulated their target genes at back skin. These results provide good insights into the inherent heterogeneity of skin SMCs in different anatomical sites.

## Heterogeneity of skin ECs in different anatomical sites

Previous studies discovered that ECs underlie the vascular systems and primarily participate in blood and skin homeostasis (*Kalucka et al., 2020*). Here, ECs were captured from six different anatomical sites and were classified into five subpopulations, which were visualized with the t-SNE plot (*Figure 4A*). GO functional terms analysis was carried out according to the enriched expression genes for each cluster, which were closely related to some functional terms, including angiogenesis, immune response, response to viruses, cell migration, cell adhesion, and regulation of catalytic activity (*Figure 4—figure supplement 1A*). To validate the spatial position of ECs in dermis, we detected the expression levels of representative PECAM1 and APOA1 genes via the immunofluorescence of skin section (*Figure 4B*).

Further, we classified EC phenotypes according to previous reported methods (*Li et al., 2021*; *Wang et al., 2022*) and found that ECs were composed of arteriole ECs expressing markers SEMA3G and MECOM (clusters 7 and 10), capillary ECs expressing marker PLVAP (cluster 3), venule ECs expressing markers SELE and ACK1 (cluster 4), and lymphatic ECs expressing markers LYVE1 and PROX1 (cluster 11) in dermis (*Figure 4C*). The pseudotime trajectory analysis of EC phenotypes showed an organized axis of blood ECs starting from arteriole and ending at venule, and it also formed an arteriovenous anastomosis tendency (*Figure 4D*). Due to exhibiting diverse molecules and functions on EC phenotypes, we further explored the expression levels of these EC phenotype-related genes, which were involved in integrins (focal adhesion), immune (cell adhesion molecules, chemokine signaling pathway, antigen processing and presentation, leukocyte transendothelial migration, and Th1 and Th2 cell differentiation), and metabolism (inositol phosphate, mucin type o-glycan biosynthesis, ether lipid, sphingolipid, and glycerolipid). ECs-related metabolism in our dataset was considerably active in arteriole ECs, especially cluster 10 involving ACER3, which controlled the homeostasis of ceramides, and LCLAT1, a lysocardiolipin acyltransferase-regulating activation of mitophagy (*Figure 4E*). The focal adhesion genes were more significantly upregulated in arteriole ECs and lymphatic ECs compared to other phenotypes, including ACTG1, BIRC3, and THBS1 (*Figure 4—figure supplement 1B*). In cell adhesion molecules, PTPRM and CDH5, mainly responsible for intercellular adhesion between ECs, were highly enriched in arteriole ECs; meanwhile, PECAM1, SELE, and SELP were enriched in venule ECs (*Figure 4—figure supplement 1C*). Other immune pathways showed that different EC phenotypes significantly highly expressed diverse genes, such as CXCL14 (involved in monocyte and recruitment) in capillary ECs, CCL26, CXCL19, and CCL26 in venule ECs (*Figure 4—figure supplement 1D*). The observed functional diversity of EC phenotypes proved the degree of ECs heterogeneity.

Here, we found that the cell number of EC phenotypes was different among different anatomical sites, with the back skin holding the most arteriole ECs and minimal lymphatic ECs (*Figure 4—figure supplement 1E and F*). To further confirm the heterogeneity of EC phenotypes in the six different anatomical sites, we compared the expression levels of these gene-related integrins, immune, and metabolism pathways (*Figure 4F* and *Figure 4—figure supplement 2A–C*). For example, for the metabolism pathway, compared to other sites, the activity of capillary ECs, venule ECs, and arteriole ECs (cluster 7 not including cluster 10) was depressed in shoulder skin, while high activity in capillary ECs, venule ECs, and arteriole ECs was shown in leg skin, including ACER3 and PIK3C2A, which enhanced cell viability (*Gulluni et al., 2021*), and high activity in lymphatic ECs, arteriole ECs (cluster 10 not including cluster 7), and venule ECs was presented in ear skin, including ACER3, GALNT10, and PIK3C2B, a member of class II PI3Ks controlling cellular proliferation, survival, and migration. The obtained abundant results on the gene expressions for EC phenotype-related pathways in different sites showed the heterogeneity of skin ECs for different anatomical sites.

To uncover the underlying molecular mechanisms driving the differential skin sites of ECs, we compared the DEGs with differentially compared groups among different anatomical sites (*Figure 4G*) and GO terms were implemented for upregulated and downregulated genes (*Figure 4H*). Enriched terms, which were related to long-chain fatty acid transport, lipoprotein particle binding, and ECM, were shown in upregulated differential genes, while downregulated differential genes mainly existed in terms involving regulation of catalytic activity, acting binding, and molecular adaptor activity. Of note, the CD36, a multifunctional fatty acid transporter, was related to the metabolic state of fibroblasts for ECM regulation (*Zhao et al., 2019*). Here, we found that CD36 was upregulated in the back compared with others, except head, which was enriched in metabolic terms such as long-chain fatty acid transport and regulation of nitric oxide. FABP4, fatty acid-binding protein 4, was significantly

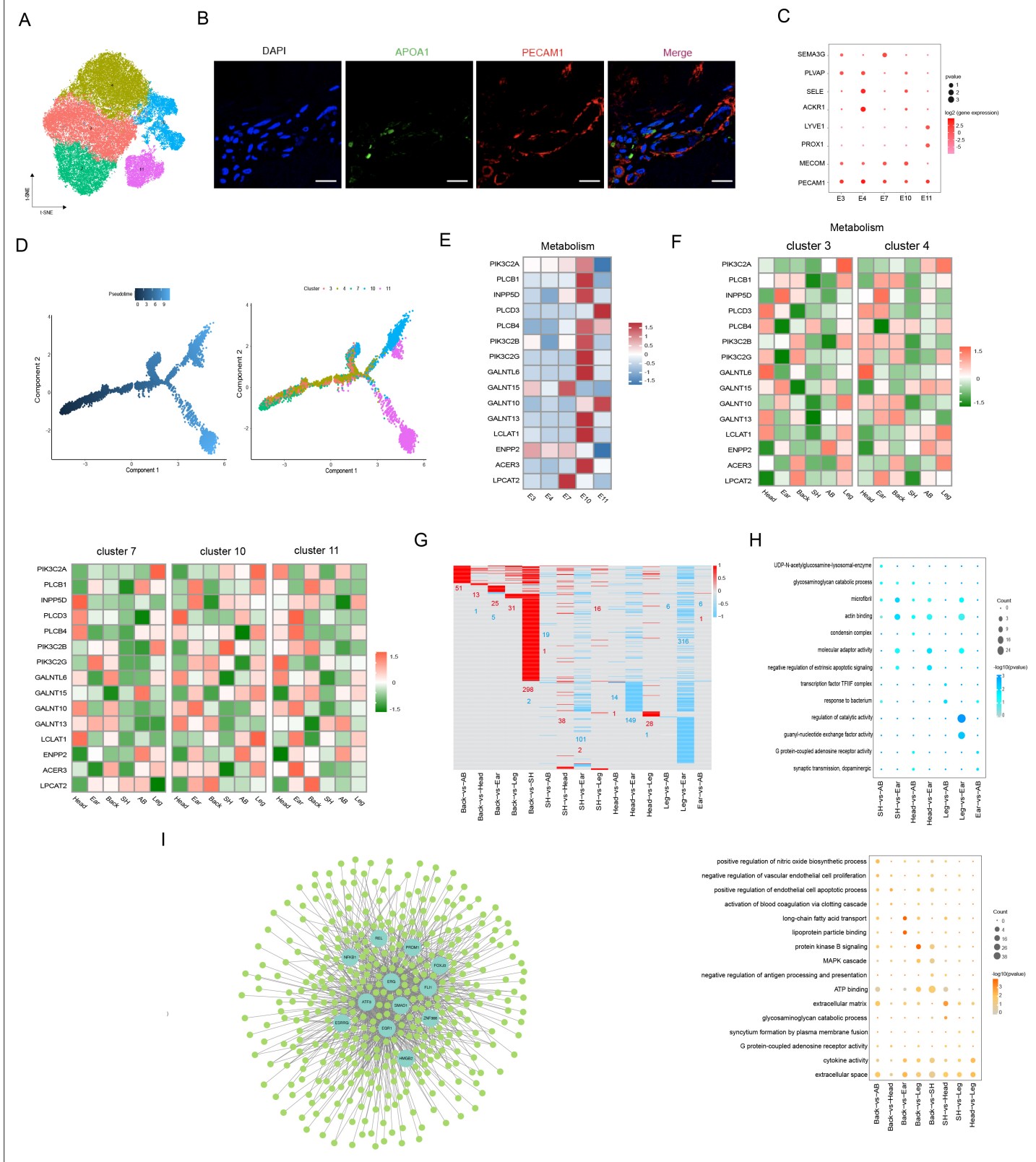

**Figure 4.** Heterogeneity of skin endothelial cells (ECs) in different anatomical sites. (**A**) The t-SNE plot visualization of ECs. (**B**) Immunofluorescence staining of APOA1 (green) and PECAM1 (red) in back skin sections, representative markers of ECs. Scale bar = 50 μm. n = 3. (**C**) Dot plot representing marker genes of ECs phenotypes. Color indicates gene expression, and circle indicates the log2FC value. (**D**) Pseudotime ordering of ECs subpopulations using monocle 2. (**E**) Heatmap showing the gene expression of metabolic pathways in ECs subpopulations. (**F**) Heatmap of gene

*Figure 4 continued on next page*

*Figure 4 continued*

expression of metabolic pathways in ECs subpopulations of different skin regions. (**G**) Heatmap of differentially expressed genes (DEGs) for ECs in multiple compared groups. Red represents upregulated genes, and blue represents downregulated genes. (**H**) The significantly enriched Gene Ontology (GO) terms of ECs in multiple compared groups. (**I**) Regulatory network of DEGs for ECs of different skin regions. Blue nodes represent regulators, and green nodes represent the target genes of regulators.

The online version of this article includes the following source data and figure supplement(s) for figure 4:

**Source data 1.** Source data of the differentially expressed genes of endothelial cells (ECs) in multiple compared groups in *Figure 4G*.

**Figure supplement 1.** Heterogeneity of skin endothelial cells (ECs) in different anatomical sites.

**Figure supplement 2.** Heterogeneity of skin endothelial cells (ECs) in different anatomical sites.

differentially expressed in nine pairs compared groups. A pioneering study showed that FABP4 was strongly expressed in subcutaneous adipocytes and adipose ECs (*Wang et al., 2022*). Combining data indicated that skin thickness might have a positive correlation with subcutaneous fat deposits. Additionally, we constructed single-cell transcription-factor regulatory networks with all DEGs of ECs (*Figure 4I*). The analysis predicted the following main transcriptional factors: ATF3, EGR1, ERG, FLI1, PRDM1, and NFKB1. The expression levels of ATF3, EGR1, and ERG were predominantly regulated in back and leg sites. With these findings, we presented the heterogeneity of skin ECs in different anatomical sites.

## Heterogeneity of skin FBs in different anatomical sites

The dermal FBs synthesize the ECM that forms the connective tissue of skin dermis to maintain the skin morphology such as thickness and homeostasis (*Zhao et al., 2019*). Phenotypic data showed that the CH skin thickness of differential anatomical sites showed striking difference such as back skin thickness on average at 5.48 mm and that of the ear at 1.52 mm (*Figure 5—figure supplement 1A*). In terms of overall skin section, the skin histomorphology of different anatomical sites exhibited some difference in sparsity of collagen fibers or the number of appendages, and dermal thickness descended from the back, head, shoulder, leg, abdomen, to the ear (*Figure 5A* and *Figure 5—figure supplement 1B*). Curiously, we inquired whether the discrepancy in ECM accumulation in different skin sites was caused by FBs heterogeneity. Next, the FBs single-cell maps were presented from six different skin anatomical sites using the t-SNE plot, which was established by four clusters (clusters 1, 8, 9, and 12) (*Figure 5B*), and the cell number of clusters was estimated (*Figure 5C*). Similar to previous reports (*Philippeos et al., 2018*; *Solé-Boldo et al., 2020*), here FBs in cluster 1 highly expressed MGP and MFAP5, known markers of reticular FBs, the most representative markers of COL6A5, WIF1, and APCDD1 of papillary FBs belonged to clusters 8 and 9, and the mesenchymal subpopulation signature was typically characterized by enriched expressed CRABP1, TNN, and SFRP1 in cluster 12. GO analysis showed that the functions of these four FBs clusters were closely related to ECM organization, collagen fibril organization, and cell adhesion (*Figure 5—figure supplement 1C*). Likewise, the label-LUM and POSTN genes were marked on FBs of skin section via immunofluorescence (*Figure 5D*).

With the discrepancy in ECM accumulation in different skin sites, we showed the expression levels of 417 upregulated or downregulated DEGs among the diverse compared groups by heatmap, in which almost all of the DEGs were upregulated in back skin compared with other skin sites (*Figure 5E*). The remarkable GO enrichment terms, including ECM, extracellular region, extracellular space, and collagen-containing ECM, were found in all compared groups (*Figure 5F*). To further explore the key genes causing the discrepancy in ECM accumulation, the top DEGs were visualized in the compared groups (*Figure 5G* and *Figure 5—figure supplement 1D*). The point photograph presented some overlapping genes in multiple compared groups, especially back skin compared with other skin sites, including TNN, COL11A1, SFRP1, COL6A5, INHBA, APOA1, IGF1, and SPARCL1. Notably, TNN, called tenascin-N(W), is a lager domain glycoprotein that has the potential to modify cell adhesion and typically contribute to cell motility (*Chiquet-Ehrismann and Tucker, 2011*); COL11A1, an ECM structural constituent, comprises a subclass of regulatory collagen fibrillogenesis that synergistically assemble other types of collagen such as collagen I, determining fibril structure, fibril organization, and functional traits (*Smith and Birk, 2012*; *Sun et al., 2020*); SFRP1, a member of secretory glycoprotein SFRP family, is regarded as one of the main classes of macromolecules making up the ECM

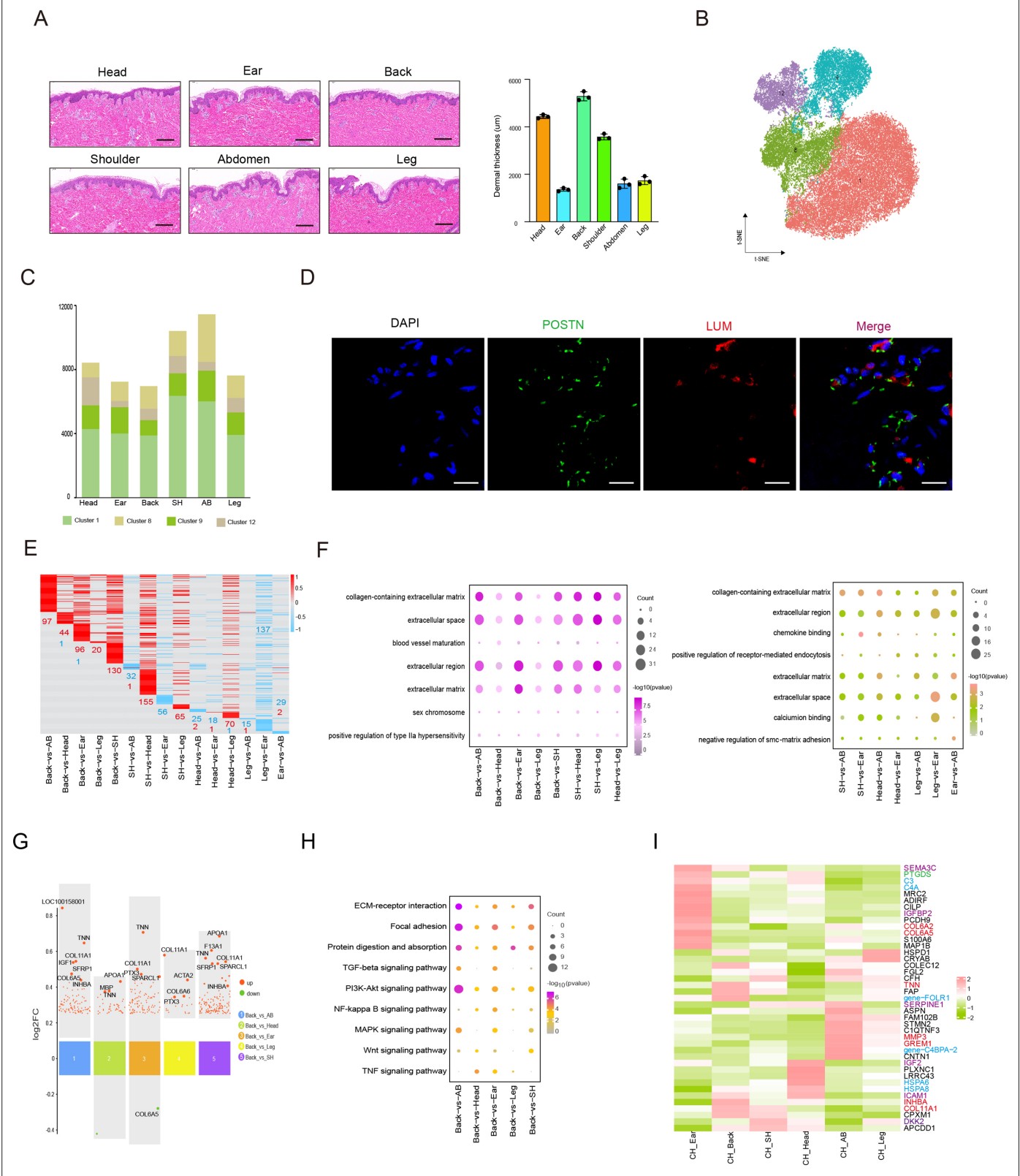

**Figure 5.** Heterogeneity of skin fibroblasts (FBs) in different anatomical sites. (**A**) Skin section with HE staining (left) and dermal thickness of six different sites (right) including head, ear, back, shoulder, abdomen, and leg. Scale bar = 100 μm. n = 3. (**B**) The t-SNE plot showing FBs populations. (**C**) The cell number of FBs populations in different skin regions. (**D**) Images showing immunofluorescence staining of POSTN (green) and LUM (red) in back skin sections, representative markers of FBs. Scale bar = 50 μm. n = 3. (**E**) Heatmap of differentially expressed genes (DEGs) for FBs in multiple compared

*Figure 5 continued on next page*

*Figure 5 continued*

groups. Red represents upregulated genes, and blue represents downregulated genes. (**F**) The enriched Gene Ontology (GO) term of FBs in multiple compared groups. (**G**) Multiple volcanic maps showing the DEGs of compared groups in back skin compared to other locations. Representative genes are indicated. (**H**) Kyoto Encyclopedia of Genes and Genomes (KEGG) analysis of representative genes in image (**G**). (**I**) Gene expression signature divides FBs from six different skin regions by unsupervised hierarchical clustering. These genes involved in extracellular matrix (ECM) synthesis (red), cell signaling guidance (purple), metabolism (green), and human diseases (blue) are labeled by the related colors.

The online version of this article includes the following source data and figure supplement(s) for figure 5:

**Source data 1.** Source data of the differentially expressed genes of fibroblasts (FBs) in multiple compared groups in *Figure 5E*.

**Figure supplement 1.** Heterogeneity of skin fibroblasts (FBs) in different anatomical sites.

**Figure supplement 2.** Heterogeneity of skin fibroblasts (FBs) in different anatomical sites.

elements and is reported to be an antagonist that inhibits human hair follicles recession (*Bertolini et al., 2021*; *Jiang et al., 2022*); INHBA is a member of TGFβ superfamily and is modified by AP1 expression (*Ham et al., 2021*).

Subsequently, we implemented KEGG analysis in the compared groups (*Figure 5—figure supplement 1E and F*) and presented dominant enrichment pathways such as ECM–receptor interaction, focal adhesion, protein digestion and adsorption, and TGF-beta signaling pathway. Interestingly, these typical overlapping genes were tightly connected with ECM production (*Figure 5H*). At a consequence, TNN, COL11A1, and INHBA were considered as key candidate genes for provoking ECM accumulation.

FBs-related gene expression signatures could serve as an important molecular cue for positional identity (*Rinn et al., 2006*), so we further revealed the significant enriched genes related to different anatomical sites from 417 DEGs among the diverse compared groups. Hierarchical clustering of four clusters for FBs was presented by the expression levels of these DEGs at six different skin anatomical sites. Hierarchical clustering of cluster 1 and 12 demonstrated that the gene expression patterns of FBs in thick sites (head, shoulder, and back) and thin sites (ear, abdomen, and leg) were partly defined by position with each structure (*Figure 5—figure supplement 2A*). Meanwhile, FBs-related gene expression patterns of clusters 7 and 8 in the upper half body sites (head, ear, and shoulder) and the lower half sites (back, abdomen, and leg) were partly defined by their anatomical position of origin. The site-sepecific genes were primarily involved in ECM synthesis, cell signaling guidance, metabolism, and human diseases (*Figure 5I*). For example, in terms of ECM synthesis, ear FBs included COL6A2 and COL6A5, abdomen FBs were involved in GREM1 and MMP3, and back FBs covered COL11A1, INHBA, and TNN. In cell signaling guidance, ear FBs included IGFBP2 and SEMA3C, abdomen FBs were involved in SERPINE1, head FBs covered ICAM1 and IGF2, and shoulder FBs referred to DKK2. In addition, The SCENIC algorithm demonstrated NF-kB1, TBX3, and ZNF366 regulons regulated some DEGs in FBs (*Figure 5—figure supplement 2B*). Of note, the targeted INHBA is targeted by TBX3 regulons. These results showed the gene expression patterns of skin FBs and site-sepecific genes in different skin anatomical sites.

## Heterogeneity of skin cells in different pig populations

Our previous studies discovered that the two pig populations, namely CH and LW pigs with a significant difference in skin thickness, showed gene expression changes at whole transcriptome levels in skin tissues and small extracellular vesicles derived from skin FBs (*Zou et al., 2022*; *Zou et al., 2023*). Curiously, the pattern of heterogeneity in skin cells is whether it also exists in the two pig populations and contributes to the difference of skin thickness. Here, we first found that the average dermal thickness of CH pigs was 5288.5 μm but only 2609.5 μm for the LW pigs (*Figure 6A* and *Figure 6—figure supplement 1A*). Then, we also implemented single-cell sequencing for the back skin of LW pigs and received a total of 18,441 cells after removing minimum count cells, doublet cells, and more than 5% cell-contained mitochondrial genes. The t-SEN analysis revealed the 18 cell clusters composed of six cell types, including SMCs (clusters 1, 2, 5, 6, 9, 10, and 14), ECs (clusters 0, 3, 4, 12, and 15), FBs (clusters 7, 8, 11, and 13), T cells (TCs) (cluster 16), MDCs (cluster 17), and ESCs (cluster 18) between CH and LW pigs (*Figure 6B*), and the marker genes of each cluster are shown in *Figure 6C*. The distribution of cell types and genes/UMIs per cell were compared between the two populations (*Figure 6D* and *Figure 6—figure supplement 1B–D*). The data showed that the main cell types were still SMCs,

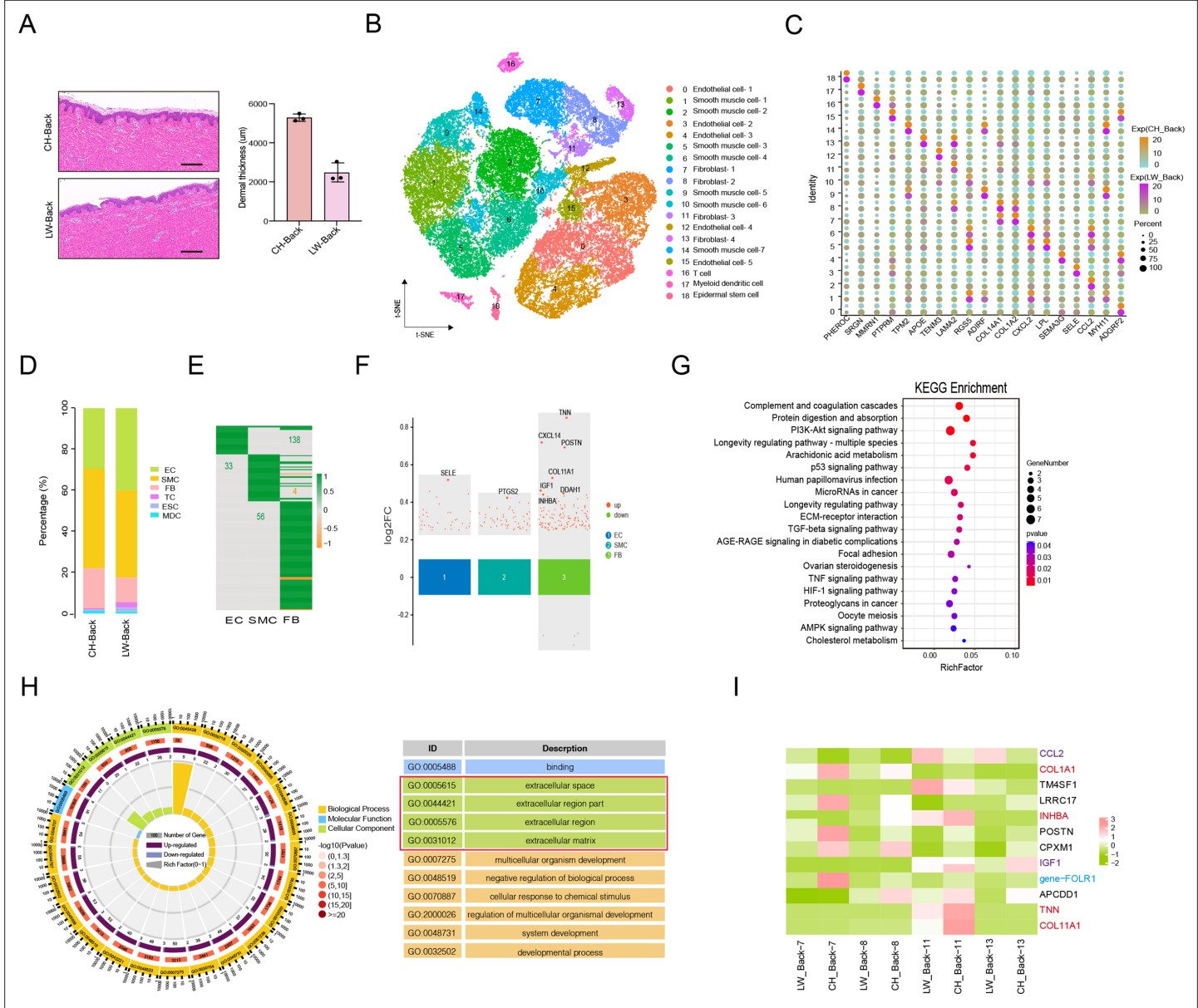

**Figure 6.** Heterogeneity of skin cells in different pig populations. (**A**) Skin section with HE staining (left) and dermal thickness of back skin between Chenghua (CH) and large white (LW) pig (right). Scale bar = 100 μm. n = 3. (**B**) The t-SNE plot visualization of all clusters of annotated cell types between CH and LW pigs. (**C**) Representative genes of each cluster of skin cells between CH and LW pigs. Color represents the gene expression, and circle represents the percentage of cells. (**D**) The distribution of cell types between CH and LW pig skin tissues. (**E**) Heatmap of differentially expressed genes (DEGs) for smooth muscle cells (SMCs), endothelial cells (ECs), and fibroblasts (FBs). Green represents upregulated genes, and orange represents downregulated genes. (**F**) Multiple volcano maps of DEGs for SMCs, ECs, and FBs. Representative genes are indicated. (**G**) Kyoto Encyclopedia of Genes and Genomes (KEGG) analysis of DEGs for FBs. (**H**) Gene Ontology (GO) term of DEGs for FBs. Red region is the most enriched GO terms. (**I**) Gene expression signature divides FBs from back skin of CH and LW pigs by unsupervised hierarchical clustering. These genes involved in ECM synthesis (red), cell signaling guidance (purple), and human diseases (blue) are labeled by the related colors.

The online version of this article includes the following source data and figure supplement(s) for figure 6:

**Source data 1.** Source data of marker genes for each cluster in *Figure 6C*.

**Source data 2.** Source data of the differentially expressed genes of smooth muscle cells (SMCs), endothelial cells (ECs), and fibroblasts (FBs) in the compared group in *Figure 6E*.

**Figure supplement 1.** Heterogeneity of skin cells in different pig populations.

ECs, and FBs in two pig populations. Meanwhile, the marker genes for SMCs, ECs, and FBs showed the most significantly enriched pathways (*Figure 6—figure supplement 1E–G*), which were consistent with that of different anatomical skin cells from CH pigs.

We further compared the DEGs in two populations (*Figure 6E and F*), which showed a significant difference in FBs. KEGG analysis for the DEGs in the main three cell types manifested significant pathways of ether lipid metabolism for ECs, including LPCAT2 and ENPP2 genes, PPAR signaling pathway for SMCs, and PI3-Akt signaling pathway, protein digestion and absorption, ECM–receptor interaction, focal adhesion, and TGF-beta signaling pathway for FBs involved in TNN, POSTN, COL11A1, IGF1, and INHBA genes (*Figure 6G* and *Figure 6—figure supplement 1H*). Moreover, the extracellular space and extracellular region part were the representative striking terms for DEGs in the FBs between the two populations by GO term analysis (*Figure 6H*). An analysis of the data proved the ECM accumulation in skin tissue was probably dependent on these overlapping genes among the anatomical regions or pig populations.

Furthermore, we analyzed the significantly enriched genes related to varietal divisions from 142 DEGs between back skin FBs of CH and LW pigs. As a result, there were a series of specific genes at back skin FBs of CH pigs, such as COL1A1, POSTN, TNN, INHBA, IGF1, and COL11A1; while TM4SF1, CCL2, and NGFR were specific genes at back skin FBs of LW pigs (*Figure 6I*). These results showed a transcriptome property of skin FBs in different pig populations.

## Signaling crosstalk among various cell types in skin

Intercellular communication plays an important role in complex tissues (*Jin et al., 2021*). Understanding cell–cell communication in skin tissue requires accurate signaling crosstalk via ligands, receptors, and their cofactors, and effective overview analysis of these signaling links. To investigate the signaling crosstalk among seven identified cell types in skin tissue, we established intercellular communication by the R package CellChat. The seven cell types were deemed as communication 'hub,' which detected 547 ligand–receptor pairs and were further categorized into 36 signaling pathways, including the COLLAGEN, LAMININ, FN1, PDGF, CCL, CXCL, MIF, and ITGB2 pathways (*Figure 7A*). Specifically, the COLLAGEN and LAMININ pathways exhibited highly abundant signaling interactions among seven cell types. Network centrality analysis of the COLLAGEN/LAMININ pathways revealed that FBs were the main source of the COLLAGEN/LAMININ ligands targeting SMCs and ESCs, which showed the COLLAGEN/LAMININ interactions were primarily paracrine way (*Figure 7B and C* and *Figure 7—figure supplement 1A*). Importantly, these results reported the elaborate relevance between FBs and SMCs with majority ligand of COL1A1 and COL1A2, receptor of CD44 and ITGA1+ITGB1 in the COLLAGEN pathway (*Figure 7—figure supplement 1B*). Likewise, the LAMININ pathway also showed an analogous phenomenon between FB and SMC populations via ligands LAMC1 and LAMB1, which were receptors of CD44 and ITGA1+ITGB1 (*Figure 7—figure supplement 1B*).

Then, we implemented a communication pattern analysis and uncovered the four patterns in outgoing secreting cells or incoming target cells (*Figure 7D and E*). Outgoing FBs signaling was identified by pattern #2, which represented multiple pathways such as COLLAGEN, LAMININ, FN1, PTN, ANGPTL, and THBS. Outgoing SMCs and ESCs signaling was characterized by pattern #4, which was included in CDH, ANGPT, and PDGF pathways. Outgoing ECs signaling was characterized by pattern #1, which was involved in PECAM1, MK, and NOTCH pathways. The pattern #3 presented CD45, IL1, and VEGF pathways for outgoing MDC and TCs signaling. For the incoming communication patterns of target cells, incoming FBs signaling was characterized by pattern #3, representing NCAM, CADM, and MPZ pathways. The incoming SMCs, KEs, and ESCs signaling was characterized by pattern #4, and that of ECs was characterized by pattern #2.

Furthermore, the signaling pathways were grouped according to their similarity in function or structure. The functional similarity grouping was classified into four groups (*Figure 7F*). Group #1 and #4, which dominantly included COLLAGEN, LAMININ, and PTN pathways, largely showed signaling from FBs to SMCs and ESCs. Group #3 dominantly drove PECAM1, CXCL, and CCL pathways, which represented the acquisition signaling pathway of ECs. The structural similarity grouping also was identified in four groups (*Figure 7—figure supplement 1C*). To further elaborately explore the communication among FB, SMC, and EC subpopulations, we analyzed the COLLAGEN/LAMININ pathway in the 14 clusters from the three cell types (*Figure 7G and H* and *Figure 7—figure supplement 1D and E*). The network centrality analysis showed that clusters 2, 5, and 6 of SMCs and cluster 7 of ECs likely

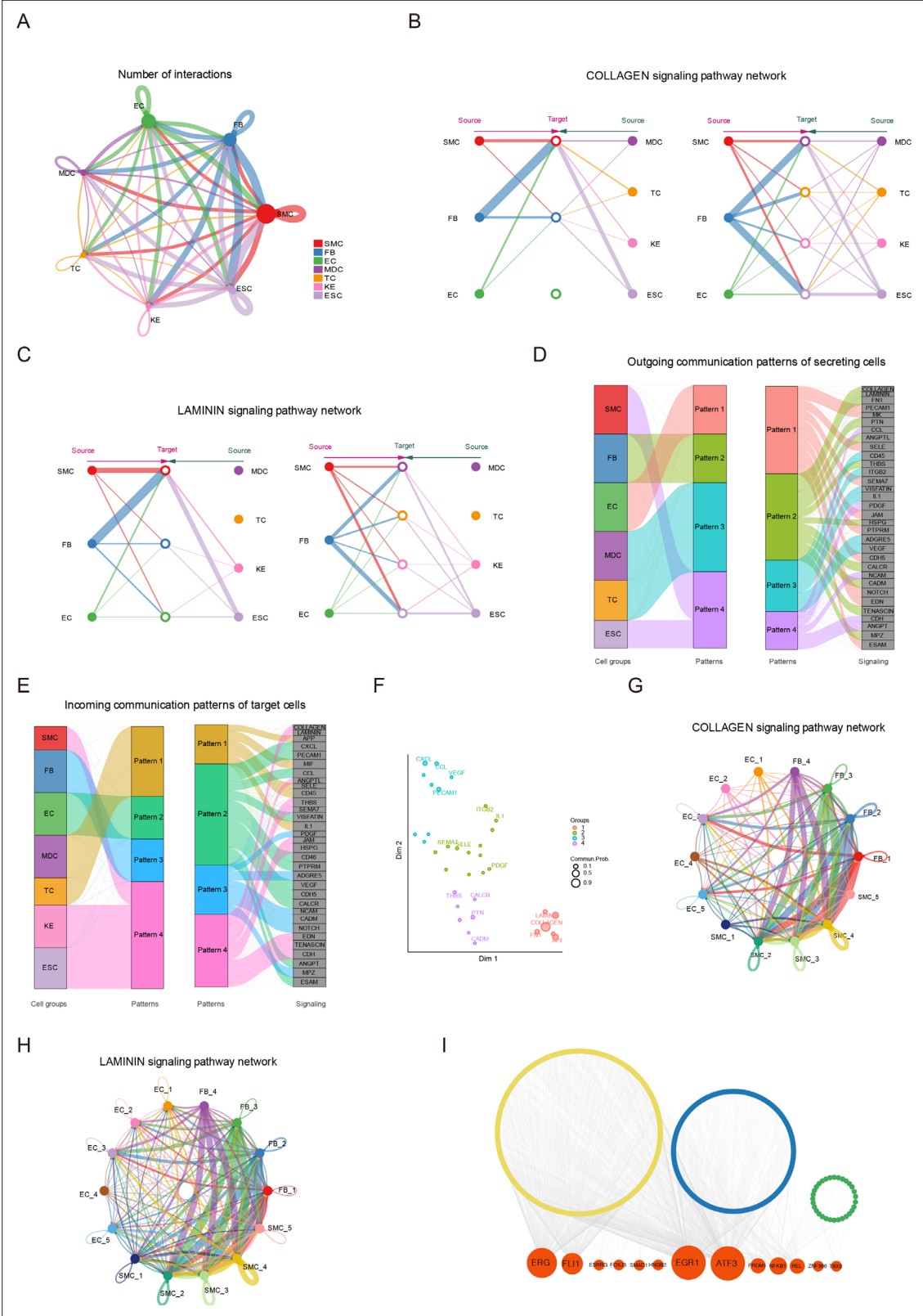

**Figure 7.** Signaling crosstalk among various cell types in skin. (**A**) Circle plot representing the cell communication among cell types. Circle sizes represent the number of cells, and edge width represents the communication probability. Hierarchical plot showing the intercellular communication network for the COLLAGEN (**B**)/LAMININ (**C**) signaling pathways. Circle sizes represent the number of cells, and edge width represents communication probability. The inferred outgoing communication patterns (**D**) and incoming communication patterns (**E**) of secreting cells of Chenghua (CH) pig skin.

*Figure 7 continued on next page*

*Figure 7 continued*

(**F**) The distribution of signaling pathways with their functional similarity. The COLLAGEN (**G**)/LAMININ (**H**) signaling network among cell subpopulations of smooth muscle cells (SMCs), fibroblasts (FBs), and endothelial cells (ECs). Circle sizes represent the number of cells, and edge width represents communication probability. (**I**) Regulatory network of differentially expressed genes (DEGs) for SMCs, ECs, and FBs of different skin regions. Orange nodes represent regulators, and yellow/blue/green nodes represent the target genes of regulators.

The online version of this article includes the following source data and figure supplement(s) for figure 7:

**Source data 1.** Cell communication of skin cells in *Figure 7A*.

**Figure supplement 1.** Signaling crosstalk among various cell types in skin.

actively take part in cell communication via the ligand of COL1A1/LAMA2, receptor of ITGA1+ITGB1 in the COLLAGEN/LAMININ pathway. Moreover, the SCENIC algorithm demonstrated NFKB1 as the common regulon among DEGs of three cell types, and the EGR1 and ATF3 regulons regulated the target genes in SMCs and ECs (*Figure 7I*). These results manifested the intercellular communication of skin cells, especially three main cell types of SMCs, ECs, and FBs.

## Discussion

With the development of high-resolution single-cell sequencing, this is applied to delineate the atlas of diverse cell types and determine the molecular basis underlying the cell heterogeneity in many complicated biological processes associated with physiology and pathology among species, especially humans (*Han et al., 2020*), mice (*Kalucka et al., 2020*), monkeys (*Han et al., 2022b*), or pigs (*Wang et al., 2022*). The origin of tissue, its development state, or anatomical position are conducive to heterogeneity of cells. Using scRNA-seq, previous studies uncovered the heterogeneity of skin cells and their gene expression changes during embryonic development (*Ge et al., 2020*), age stages (*Zou et al., 2021*), and wound-healing stages (*Guerrero-Juarez et al., 2019*) in humans or mice. However, the single-cell transcriptional diversity of different anatomical skin regions has not been understood. The CH pig, a novel Chinese indigenous breed with superior skin thickness, is considered as a potential model animal for researching mammalian skin biology (*Zou et al., 2022*; *Zou et al., 2023*).

Some studies have identified various cell types in the physiological skin tissue in humans (*Solé-Boldo et al., 2020*) and mice (*Guerrero-Juarez et al., 2019*), and the top three main cell types were FBs, KEs, and ESCs for humans, and FBs, myeloid cells, and ECs for mice. In this study, we identified seven skin cell types from six different anatomical sites and six skin cell types from different two populations in adult pigs, and the top three main cell types were SMCs, ECs, and FBs. Compared with reported skin cell types of young pigs with the top three cell types being FBs, ESCs, and KEs, the distribution ratio of cell types was significantly different (*Han et al., 2022a*). Similarly, melanocytes, Schwann cells, mast cells, and neural cells were not identified in our datasets, while they were identified in human or mouse skin samples (*Guerrero-Juarez et al., 2019*; *Zou et al., 2021*). By the way, SMCs and pericytes, called mural cells in vessels, were unable to precisely discriminate between the two cell types because of confusable hallmarks and functions in skin tissues (*Muhl et al., 2020*; *Han et al., 2022b*), so we only identified SMCs in our data. The discrepancy in captured cell types and cell-type proportions might come from the heterogeneity of species, development process, skin state, and different scRNA-seq platform captures.

SMCs constitute blood vessels and APM in skin tissue, with a greater proportion in blood vessels (*Driskell et al., 2013*; *Muhl et al., 2020*). Here, five SMC subpopulations are verified and then separated into three SMC phenotypes in pig skin tissue. Sophisticated studies have shown SMC phenotypic switching under pathological processes or injured conditions, a way in which SMC shift between contractile phenotype and other type cell phenotypes such as mesenchymal-like, fibroblast-like, macrophage-like, adipocyte-like, and osteogenic-like (*Yap et al., 2021*; *Yu et al., 2022*). In addition, several studies found multiple SMC phenotypes, including Scal-positive vascular SMC-lineage, also existed in healthy vessels (*Dobnikar et al., 2018*). Interestingly, we found that SMC phenotypic switching also occurred in pig skin tissue, varying from contractile SMCs to mesenchymal-like, mesenchymal-like to macrophage-like, with the expression levels of marker genes for cell types and function analysis. During physiological and pathological angiogenesis, macrophages are regarded as a facilitator of vascular integrity and derivatives by way of cytokine secretion and ECM remodeling (*Barnett et al., 2016*; *Debels et al., 2013*), which implies SMCs are deemed immune system's line of

defense and positively participate in the immune response in skin tissue. Besides, in regard to different anatomical skin regions, we found that the cell number of the macrophage-like phenotype is highest at back site, followed by the abdomen, but immune-related genes primarily existed in shoulder, back, and ear, which indicates that the activity of the macrophage-like phenotype might depend on intrinsic factors as well as environmental factors.

Depending on the properties of diverse molecular and function, such as immune responses and metabolic process in ECs, ECs' heterogeneous characteristics have been investigated in some organs of human skin (*Li et al., 2021*), the mouse brain (*Kalucka et al., 2020*), and pig adipose tissue (*Wang et al., 2022*), of which ECs' diversity remain largely unrevealed in skin tissue from different anatomical locations. In our cutaneous datasets, five subpopulations are identified with ECs and divide into four EC phenotypes, which are placed in order at the pseudotime trajectory indicating the distributed paths of blood vessel, such as arteriovenous anastomosis with vein. Here, an additional level of heterogeneity was explored when analyzing the expression levels of pathway genes involved in integrins, immune, and metabolism in EC subtypes from different anatomical regions. In integrins, ITGA6 is highly expressed in pig dermis ECs, in accordance with human dermis ECs (*Li et al., 2021*), and the expression of ITGA6 is significantly added in venule ECs and arteriole ECs (cluster 10 not cluster 7) of the pig ear skin site and in capillary ECs of humans. Cell adhesion molecules are compared to find capillary ECs and arteriole ECs (cluster 7 not cluster 10) of the pig back and shoulder skin mainly enriched MHC class II genes such as SLA-DQB1 and SLA-DRB1, which were highly expressed in lung organ of humans/mice, indicating a role in immune surveillance (*Goveia et al., 2020*). A funny question regarding the high expression of MHC class II genes in terms of slight tissue rejection by blocking MHC class II on human endothelium (*Abrahimi et al., 2016*) is whether there is a preference for skin graft from specific skin regions to transplant pig skin into humans.

SELE, SELP, and ICAM1 mainly mediate the communication between leukocyte and ECs and are highly expressed in venule ECs and arteriole ECs of head, ear, and back skin of pig, while the three genes are primarily existed in post-capillary venule ECs of human dermis (*Li et al., 2021*). CDH5, an intercellular tight junction protein in ECs, is highly expressed in arteriole ECs of the ear skin of pig, in keeping with human dermal EC phenotype. Other immune response representative genes such as cytokines (CXCL14, CCL26, CCL24, CXCL12, and CXCL19) that participate in immunocyte recruitment (e.g., neutrophils) or are responsible for the host defense against viral infection, enhancing immune progression and metastasis (*Fajgenbaum and June, 2020*; *Wu et al., 2020*), of which are CCL24/CCL26, the role of eotaxins (*Provost et al., 2013*), are enriched in venule ECs and majorly distributed in the head/abdomen skin regions, respectively. For the ECs metabolism pathway, most metabolic genes are significantly expressed in arteriole ECs and exhibited overlapping and were specific among different skin sites. Interestingly, ENPP2 of lipid metabolism gene was reported to enhance the cytokine production (*Grzes et al., 2021*) and was overexpressed during chronic inflammation (*Argaud et al., 2019*), and it was enriched in the abdomen skin, while LPCAT2 of the other lipid metabolism gene under study was positively correlated with lipid droplet content in colorectal cancer (*Cotte et al., 2018*), which was mainly highly expressed in back and shoulder skin. These findings demonstrated the extensive phenotypic plasticity and gene expression signatures of all kinds of pathways in different skin sites.

FBs are mesenchymal cells that synthesize ECM of connective tissues, which are responsible for structural integrity, wound repair, and fibrosis in skin (*Driskell and Watt, 2015*). Providing plentiful proofs showed that FBs heterogeneity is involved in diverse subpopulations such as papillary FBs, reticular FBs, mesenchymal FBs, and pro-inflammatory FBs, and its functions in humans and mice (*Guerrero-Juarez et al., 2019*; *Zou et al., 2021*). In this study, we identified three subpopulations not including pro-inflammatory in different anatomical sites and populations, guessing the immune function of SMCs or ECs might replace pro-inflammation FBs due to the enormous cell number of SMCs or ECs. A previous study showed that FBs from distinct anatomical locations exhibited detectable differences in metabolic activity (*Castor et al., 1962*) and genome-wide gene expression changes in 43 skin sites (*Rinn et al., 2006*). The fact is that three FB subpopulations focused on ECM organization and collagen fibril organization, resulting in the discrepancy in ECM deposition in different anatomical skin sites and populations.

Therefore, the overlapping remarkable upregulate genes were found among multiple compared groups, especially the back when compared with other areas, and they might be regarded as key

genes in ECM deposition. The ECM protein TNN is highly expressed in dense connective tissue such as cartilage, adult skeleton, and bone (*Chiquet-Ehrismann and Tucker, 2011*). TNN, distinctly located with collagen 3 fibers, plays a crucial role in periodontal remodeling; an example of a dense scar-like connective tissue enriched the nerve fibers replacing alveolar bone around the incisor by deficient TNN in mice (*Imhof et al., 2020*). Here, TNN took part in these pathways that were closely related to skin dermis such as ECM–receptor interaction and PIK-Akt signaling pathway and were significantly upregulated in multiple compared groups uniformly, surmising that TNN might be a key candidate gene for ECM deposition. Collagen I is the most abundant structural macromolecule in skin tissue, and collagen mechanism is determined by a minor component as a regulator (*Hansen and Bruckner, 2003*). Collagens I and XI can package into composite fibrils by nucleation and propagation, in which the collagen XI content is closely connected with collagen I, determining its organization and function properties. Collagen XI is the main factor in collagen I-containing tissue, including tendons and cartilage, but not skin tissue, and the absence of COL11A1 expression results in the disruption of fibril phenotype for mature tendons (*Blaschke et al., 2000*; *Sun et al., 2020*). INHBA plays an important role in the TGF-beta signaling pathway, stimulating the activity of SMAD2/3 and encouraging cell proliferation and ECM production. INHBA expression was significantly upregulated in keloid FBs compared to normal dermal FBs (*Ham et al., 2021*).

Interestingly, overlapping genes, including TNN, COL11A1, SFRP1, and INHBA, were pronouncedly expressed in mesenchymal FBs. With the paradigm of human skin case presented, a series of genes were significantly increased in keloid mesenchymal FBs in contrast to normal scar, such as COL11A1, SFRP1, TNC, INHBA, FN1, IGF1, THBS4, and POSTN, suggesting that these genes might promote ECM production (*Deng et al., 2021*). Likewise, these TNN, POSTN, COL11A1, IGF1, and INHBA genes were significantly upregulated in the back skin of CH pig compared with the back skin of LW pig. Although the mechanisms of physiological skin thickness, fibrosis, or scarring (pathological chronic inflammatory) are not all the same, excess ECM accumulation occurs, indicating individual and mutual genes. Therefore, in our study, we speculate that TNN, COL11A1, and INHBA expression might play a critical role in the morphology and quantity of collagen fibril-stimulated ECM deposition in skin tissue.

A previous study discovered that the Hox genes are a family of TF-encoding genes that are crucial for regulating embryonic development and tissue morphogenesis (*Rinn et al., 2006*). However, in our study, we did not identify any Hox gene among these differentially expressed genes in skin fibroblasts from both different anatomical sites and different pig populations. The differences of Hox code expression patterns might come from the heterogeneity of different species. Coincidently, some signaling pathways, which are related to regional- or populations-specific genes of fibroblasts, were identified in our study, which partially match what Rinn et al. published on human skin. For example, COL11A1 and MMP3 facilitate ECM accumulation, while DDK2 and GREM are involved in cell signaling guidance, etc. These findings provided important insights into the positional identity in skin tissue.

Skin physiological and pathological (wound healing or fibrosis) conditions not only determine the complex and diverse cellular composition but also establish the central signaling pathways between interacting cell groups, offering good insights into cellular crosstalk. For mouse skin wound tissue, network analysis was categorized into 25 signaling pathways involving in TGFβ, non-canonical WNT, TNF, SPP1, and CXCL and identified the inferred TGFβ signaling as the most prominent pathway between myeloid cells and FBs (*Guerrero-Juarez et al., 2019*). Twenty-two signaling pathways of embryonic mouse skin were identified, such as WNT, ncWNT, TGFβ, PDGF, NGF, FGF, and SEMA3, predicting the WNT signaling pathway played an important role between epidermal to dermal cells to form skin morphogenesis (*Gupta et al., 2019*). Moreover, the major highly active pathways in diseased human skin included MIF, CXCL, GALECTIN, FGF, and CCL, which showed that MIF signaling pathway was the main pathway from inflammatory FBs to inflammatory TCs (*He et al., 2020*). In our datasets, 36 signaling pathways were shown to be involved in COLLAGEN, LAMININ, FN1, PDGF, CCL, CXCL, and MIF, of which the COLLAGEN and LAMININ signaling were the most enriched among different skin regions of pig. These results indicate the key signaling pathways depended on skin morphogenesis.

In summary, in our study, the heterogeneity of main cell types from both different anatomical skin sites and different pig populations was comprehensively detailed, giving clear evidence of the use of

pig as an excellent skin model focused on generation, transmission, positional information, and transplant, paving the foundation for skin priming.

## Materials and methods

### Skin samples dissociation and cell collection

Skin samples were obtained from three female 180-day-old CH pigs at six different anatomical body areas (head, ear, shoulder, back, abdomen, and leg) and three female 180-day-old LW pigs with one region (back). The fresh full skin samples were thoroughly scraped off the hair and subcutaneous fat and were washed thrice with ice-cold Dulbecco's phosphate-buffered saline (1×DPBS). The skin samples (size approximately 2 cm × 2 cm) were fully dissected into small pieces in 4 mL tube and then transferred into 50 mL centrifuge tube with 15 mL mix digestion medium containing 1 mg/mL collagenase type I, II, IV, and V (Sigma-Aldrich, Saint Louis), 1 mg/mL elastase (Coolaber, Beijing, China), and 2 U/mL DNase I (Coolaber) in Dulbecco's Modified Eagle Medium (DMED). The skin samples were digested at 37°C for 120–180 min and simultaneously gently shaken once every 10 min. The digestion reaction was interrupted by DMEM including 10% fetal bovine serum (FBS) (Gibco, New York). Then, the tissue suspension was filtered with 70 μm and 40 μm cell strainer and transfected into a 15 mL centrifuge tube to obtain cell sediment by centrifugation at 350 × $g$ for 5 min at 4°C until there was no undissociated tissue debris. The cell sediment was added to 2 mL Red Blood Cells Lysis Solution (QIAGEN, Duesseldorf, Germany) at room temperature for 5 min to remove red blood cells. The cell sediment was added to 2 mL TrypLE (Gibco) at 37°C for 45 min to dissolve cell clot. The dissociated cells were washed twice and resuspended in cold DMED supplemented with 10% FBS. Finally, cells staining with 0.4% Trypan Blue Solution was used to estimate cell activity rate and concentration by Countess Cell Counting Chamber Slides.

### Single-cell library construction and sequencing

Approximately 20,000 cells were captured in droplet emulsions, and the mRNA of single-cell libraries were constructed according to the DNBelab C Series Single-Cell Library Prep Set (MGI, Shenzhen, China) (*Han et al., 2022a*). In brief, single-cell suspensions were subjected to a series of progress, including droplet encapsulation, emulsion breakage, mRNA captured bead collection, reverse transcription, and cDNA amplification and purification, to generate barcoded libraries. Indexed sequencing libraries were established based on the instruction's protocol. The quality supervision of libraries was implemented with a Qubit ssDNA Assay Kit (Thermo Fisher Scientific, Waltham). Libraries were further sequenced by the DNBSEQ sequencing platform at the China National GeneBank.

### Single-cell RNA sequencing data processing

The raw single-cell sequencing data were processed by DNBelab C Series scRNA analysis software. Reads were aligned to the reference genome (Ensemble assembly: Sus scrofa11.1) to generate a digital gene expression matrix by STAR (*Wang et al., 2022*). The quality control parameters involved in gene counts per cell, UMI count per cell, and % mitochondrial genes were stipulated. Cells genes were expressed in less than three cells, and cells were removed on the basis of detected genes number with a minimum of 200. Mitochondrial gene expression was set at a threshold of 5% per cell. For each library, the doublet was removed using DoubleFinder with the default parameter (*Wang et al., 2022*). Then, the aligned reads were filtered to obtain cell barcodes and UMI for gene-cell matrices, which were used for downstream analysis.

### Identification of cell clustering and cell type

After the initial DNBelab C Series scRNA analysis software processing, the cells were preprocessed and filtered. The data were normalized per sample using NormalizaData with default options, and highly variable genes were calculated by FindVariableFeatures and then elected based on their average expression and dispersion. The cell cluster was presented with the standard integration process of p-value<0.01 through the 'FindClusters' function described in Seurat (*Wang et al., 2022*). The cell types in each cell cluster were identified with enriched expression using 'FindAllMarkers' function in SCSA with default parameters, together with canonical cell-type markers from extensive reported

literature on pig and human skin. Genes with |log2FC| > 0.25 and adjusted p-value <0.05 were considered marker genes. Subsequently, the cell cluster was visualized with t-SNE plot.

## Identification of DEGs among multiple compared groups and GO/KEGG enrichment analysis

We used the FindMarkers function in Seurat to confirm skin-related DEGs between CH-back and CH-head, CH-back and CH-ear, CH-back and CH-shoulder, CH-back and CH-abdomen, CH-back and CH-leg, CH-shoulder and CH-head, CH-shoulder and ear, CH-shoulder and CH-abdomen, CH-shoulder and CH-leg, CH-head and CH-ear, CH-head and CH-abdomen, CH-head and CH-leg, CH-leg and CH-abdomen, CH-leg and CH-ear, and CH-abdomen and CH-ear for each cluster. DEGs of 15 compared groups were identified with |log2FC| > 0.25 and adjusted p-value <0.05. In the global clusters, GO analysis was implemented with the Dr. Tom platform of BGI. KEGG analysis, which was also performed with the Dr. Tom platform of BGI, further identified gene biological function, including signal transduction pathways, metabolic pathways, and so on in dermal cell populations.

## Cross-species comparison for skin cell atlas in pigs, humans, and mice

Published skin single-cell datasets of humans (*Solé-Boldo et al., 2020*; *Zou et al., 2021*) and mice (*Joost et al., 2020*; *Ko et al., 2022*) were download from GEO with a 10X sequencing platform. The count matrices of the three species were integrated for clustering using the Seurat R package with standard process for interspecies skin cell atlas analysis. The expressed genes that were orthologous were kept in the three species. The comparison of cell numbers and UMI counts matrices was obtained for pigs, humans, and mice. And the cell types were annotated by cell-type marker genes identified in this study.

## Pseudotime analysis

The cell pseudotime trajectory was constructed using R package Monocle2 (*Trapnell et al., 2014*). This method arranges these cells on a trajectory that describes the complete differentiation process as a quasi-time sequence of these cells through the asynchronous nature of each cell in the differentiation process.

## Cell–cell communication inference

To understand global communication among the cell types of pig skin, we used the R package Cell-Chat (v1.0.5) (*Trapnell et al., 2014*) with ligand–receptor interactions for visual intercellular communications from scRNA-seq data. As the database covers the human species, we select these pig genes according to their homologous with humans. CellChat implements some visualization methods, including the interaction number, interaction weight, communication patterns of incoming river plot, communication patterns of outgoing river plot, functional pathways, structural pathways, chord plot, circle plot, hierarchy plot, and ligand–receptor of contributions.

## Targeted transcription factors interaction among cells

The TF list for pig species was downloaded from the AnimalTFDB (v4.0). We identified all the TFs using motif enrichment data in cisTarget database (https://resources.aertslab.org/cistarget/), of which the 'grn' module constructed a co-expression network, the 'cxt' module inferred regulome, and the 'aucell' module calculated the AUC value in SCENIS (v0.11.2) (*Kalucka et al., 2020*). From the above data, we selected the DEGs of SMCs, ECs, and FBs corresponding to TF and visualized these networks using Cytoscape software.

## Skin section

The total skin thickness from three pigs per breed with different sites was measured three times with a Vernier caliper at the same position and recorded. The skin tissues were fixed in a solution of 10% neutral buffered formalin and processed using routine histological procedures. Then, the sections were cut at a thickness of 5 µm using a microtome. The dermal thickness was determined using Case-Viewer software according to a previous method after hematoxylin-eosin staining (*Zou et al., 2022*). The mean values and standard deviations were calculated.

## Immunofluorescence staining

A 5-µm-thick back skin section was incubated with primary polyclonal rabbit antibody (ABclonal, Wuhan, China) against MYH11 (1:500 dilution) and ACTA2 (1:500 dilution) overnight at 4°C for SMCs, APOA1 (1:500 dilution) and PECAM1 (1:200 dilution) for ECs, and LUM (1:200 dilution) and POSTN (1:200 dilution) for FBs. FITC-goat anti-rabbit IgG and Cy3-conjugated goat anti-rabbit IgG were used as secondary antibodies (1:200 dilution) at room temperature for 1 hr. Then, the cell nuclei were stained with DAPI dye for 30 min. These procedures were implemented under dark conditions. Finally, these images were captured by confocal microscopy.

## Statistical analysis

Statistical testing was applied by GraphPad Prism. The data are shown as the mean ± SD for one group.

## Acknowledgements

This work was supported by the Chengdu Livestock and Poultry Genetic Resources Protection Center (2022) and Sichuan Science and Technology Program (2021ZDZX0008).

# Additional information

### Funding

| Funder | Grant reference number | Author |
|---|---|---|
| Chengdu Livestock and Poultry Genetic Resources Protection Center | 2022 | Yanzhi Jiang |
| Sichuan Science and Technology Program | 2021ZDZX0008 | Yanzhi Jiang |

The funders had no role in study design, data collection and interpretation, or the decision to submit the work for publication.

### Author contributions

Qin Zou, Conceptualization, Data curation, Investigation, Methodology, Writing – original draft; Rong Yuan, Investigation, Methodology; Yu Zhang, Data curation; Yifei Wang, Ting Zheng, Rui Shi, Mei Zhang, Methodology; Yujing Li, Kaixin Fei, Ran Feng, Binyun Pan, Xinyue Zhang, Zhengyin Gong, Investigation; Li Zhu, Guoqing Tang, Mingzhou Li, Xuewei Li, Writing – review and editing; Yanzhi Jiang, Conceptualization, Supervision, Funding acquisition, Writing – review and editing

### Author ORCIDs

Yanzhi Jiang http://orcid.org/0000-0002-9568-557X

### Ethics

Three heads per breed aged 180 days old of both CH and LW pigs were obtained from Chengdu Livestock and Poultry Genetic Resources Protection Center. All animal experimental procedures were permitted following the Care and Use Committee of Sichuan Agricultural University (permit number: 20220219).

### Decision letter and Author response

Decision letter https://doi.org/10.7554/eLife.86504.sa1
Author response https://doi.org/10.7554/eLife.86504.sa2

# Additional files

### Supplementary files
• MDAR checklist

## Data availability

The single-cell RNA-seq data have been deposited in NCBI's Gene Expression Omnibus database and accessible through GEO Series accession number GSE225416. All data generated or analyzed during this study are included in the manuscript. Source data files have been provided for Figure 1, Figure 1—figure supplement, Figure 3, Figure 4, Figure 5, Figure 6 and Figure 7.

The following dataset was generated:

| Author(s) | Year | Dataset title | Dataset URL | Database and Identifier |
|---|---|---|---|---|
| Zou Q, Yuan R, Zhang Y | 2023 | Single-cell transcriptome analysis on the anatomic positional heterogeneity of pig skin | https://www.ncbi.nlm.nih.gov/geo/query/acc.cgi?acc=GSE225416 | NCBI Gene Expression Omnibus, GSE225416 |

The following previously published datasets were used:

| Author(s) | Year | Dataset title | Dataset URL | Database and Identifier |
|---|---|---|---|---|
| Solé-Boldo L, Raddatz G | 2020 | Single-cell transcripttomes of the human skin reveal age-related loss of fibroblast priming | https://www.ncbi.nlm.nih.gov/geo/query/acc.cgi?acc=GSE130973 | NCBI Gene Expression Omnibus, GSE130973 |
| Zou Z, Long X | 2021 | A Single-Cell Transcriptomic Atlas of Human Skin Aging | https://ngdc.cncb.ac.cn/gsa-human/browse/HRA000395 | Genome Sequence Archive, HRA000395 |
| Meriet JJ, Ko KL | 2022 | NF-kB perturbation reveals unique immuneomeodulatory functions in Prx1+ fibroblasts that promote development of atopic dermatitis | https://www.ncbi.nlm.nih.gov/geo/query/acc.cgi?acc=GSE172226 | NCBI Gene Expression Omnibus, GSE172226 |
| Joost S, Annusver K | 2020 | The Molecular Anatomy of Mouse Skin during Hair Growth and Rest | https://www.ncbi.nlm.nih.gov/geo/query/acc.cgi?acc=GSE129218 | NCBI Gene Expression Omnibus, GSE129218 |

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
