## [Editor Report]

This valuable manuscript provides a single-cell RNA sequencing analysis of adult pig skin from different species and anatomical regions. The evidence supporting the conclusions is compelling, with identification of molecular and cellular differences in pig skin, including analysis of regional and species-specific gene signatures.

---

## [Decision Letter]

**Decision letter after peer review:**

Thank you for submitting your article "Single-cell transcriptome analysis on the anatomic positional heterogeneity of pig skin" for consideration by *eLife*. Your article has been reviewed by 2 peer reviewers, and the evaluation has been overseen by a Reviewing Editor and Marianne Bronner as the Senior Editor. The reviewers have opted to remain anonymous.

Essential revisions:

In general, the reviewers were enthusiastic about this manuscript. They were particularly interested in these data as a resource for pig skin. However, to accomplish this goal, the authors should analyze their data more extensively.

1) The reviewers thought the different body regions would be of interest as they could reveal aspects of positional identity. Specific analyses: Were there differences in the Hox code expression patterns? Did this match what Rinn et al. published on human skin? What were the pathways involved in regulating the phenotypic differences between the tissues? Perhaps a focused study utilizing the data would improve the manuscript.

2) The authors should do more analysis of the comparison between the Chenghua and Large White. Could the authors expand on the mechanisms of why the dermal thickness would be different between the two species? What inherently regulates this difference at a transcriptional level? Also, are there positional identity differences between the two species?

3) Overall the manuscript could also use editorial help in teasing out the central message of the manuscript. Is this just a resource manuscript allowing for an interesting and important dataset to be shared with the world? If so, the authors should generate a web tool.

*Reviewer #1 (Recommendations for the authors):*

Some of my suggestions are below and hope they are helpful.

1) A huge amount of effort and expense went into creating a dataset from 6 different body sites. It is extremely hard for this reviewer to ascertain from both the writing in the manuscript and the figures what the dataset says about positional identity. Were there differences in the Hox code expression patterns? Did this match what Rinn et al. published on human skin? What were the pathways involved in regulating the phenotypic differences between the tissues? Perhaps a focused study utilizing the data would improve the manuscript.

2) The comparison between the Chenghua and Large White is very interesting. The authors note that the thickness of the dermis is different and reveal an analysis that suggests that the expansion of the extra-cellular matrix genes occur in the pig with thicker skin. This is an obvious result. Could the authors expand on the mechanisms of why this would happen? What inherently regulates this difference at a transcriptional level?

3) Overall the manuscript could also use editorial help in teasing out the central message of the manuscript. Is this just a resource manuscript allowing for an interesting and important dataset to be shared with the world? If so, the authors should generate a web tool.

4) It would be interesting to create a narrative around the idea that the Chenghua and Large White have a difference in the way that positional identity is regulated between the different species. Is it possible that the authors could show this through their single-cell data?

*Reviewer #2 (Recommendations for the authors):*

1. I did not find the age and sex of the pigs. This basic information should be given.

2. Which skin layer is dissected to separate the specimen from the body? This may lead to different thickness of the skin from different regions. Are all the components dissociated into single cells? Do some cells remain undissociated in the tissue debris? All these may affect the analyses of cell compositions. These differences may be unavoidable but the condition should be clarified. Showing photos of specimens in supplements may help.

3. English should be improved. There are also typos. For example, p. 52. Pseudotiom should be pseudotime.

3. Panel 2E needs more explanation.

4. Figure 5A and 6A. H&E staining needs improvement.

5. A major possibility to improve the significance is to evaluate the functional significance further. It may not be practical for the authors. The current work is still valuable but will have limitations in its impact.

---

## [Author Response]

Essential revisions:In general, the reviewers were enthusiastic about this manuscript. They were particularly interested in these data as a resource for pig skin. However, to accomplish this goal, the authors should analyze their data more extensively.1) The reviewers thought the different body regions would be of interest as they could reveal aspects of positional identity. Specific analyses: Were there differences in the Hox code expression patterns? Did this match what Rinn et al. published on human skin? What were the pathways involved in regulating the phenotypic differences between the tissues? Perhaps a focused study utilizing the data would improve the manuscript.

(1) Hox genes are a family of transcription factor-encoding genes that are crucial for regulating embryonic development and tissue morphogenesis. However, in our study, we did not identify any Hox gene among these differentially expressed genes in skin fibroblasts from both different anatomical sites and different pig populations. The differences of Hox code expression patterns might come from the heterogeneity of different species. Coincidently, some signaling pathways, which related to regional- or populations-specific genes of fibroblasts, were identified in our study, which partially match what Rinn et al. published on human skin. For example, COL11A1 and *MMP3* facilitate ECM accumulation, while DDK2 and GREM are involved in cell signaling guidance, etc. The content have added in the Result Section.

(2) Moreover, we have further analyzed some valuable data more extensively as follows:

To ascertain the positional identity, in Result Section of “Heterogeneity of skin FBs in different anatomic sites”, we supplemented the gene expression signatures of fibroblasts among different anatomic skin sites, and these identified genes were mainly involved in ECM synthesis, cell signaling guidance, metabolism and human diseases pathways. The added results are shown in Figure 5I and Figure 5—figure supplement 2A. Meanwhile, to ascertain the population identity, we have supplemented the single-cell data for SMCs, ECs and FBs between Chenghua and Large White pigs in Result Section of “Heterogeneity of skin cells in different pig populations”, and these added data are shown in Figure 6—figure supplement 1E-G. Moreover, the FBs-related gene expression levels of specific-population between the different populations have also been added and shown in Figure 6I.

2) The authors should do more analysis of the comparison between the Chenghua and Large White. Could the authors expand on the mechanisms of why the dermal thickness would be different between the two species? What inherently regulates this difference at a transcriptional level? Also, are there positional identity differences between the two species?

We think that the artificial selection and specific feeding environment might contribute to the phenotypic formation of superior skin thickness trait in CH pigs. Moreover, previous study discovered that the dermal collagenous ECM determined individual skin thickness (Qin et al., 2018), so the inherent mechanism might result in the expansion of the extra-cellular matrix genes occur in the CH pigs with thicker skin. In fact, gene expression changes at transcription levels are influenced by multiple regulatory mechanisms, such as transcription factors, epigenetic regulation (including DNA methylation, histone modification, non-coding RNA regulation, etc.), and post-transcriptional regulation of RNA. In our previous studies, we elucidated that the difference of skin thickness between Chenghua and Large White pigs was influenced by the circ004463-miR23b-CADM3/MAP4K4 regulatory axis or miR-218 (Zou et al., 2022, 2023), which belong to non-coding RNA epigenetic regulation. In this study, we found that some transcription factor expansion of the extra-cellular matrix genes occurred in the CH pig with thicker skin and it might contribute to the difference of skin thickness between CH and LW pigs. The mechanism explanations have been supplemented and added in Result Section of “Heterogeneity of skin cells in different pig populations”.

3) Overall the manuscript could also use editorial help in teasing out the central message of the manuscript. Is this just a resource manuscript allowing for an interesting and important dataset to be shared with the world? If so, the authors should generate a web tool.

In fact, our obtained single-cell data have been uploaded to the GEO database (GEO number: GSE225416) and it is free and available for readers. Although generating a web tool is a nice suggestion, it is difficult for us to achieve it under the present conditions. Because generating a web tool will need much time (about 6 months), money (about 20,000 dollars), and professional technical help. Fortunately, we are now performing single-cell RNA sequencing on CH pig skin during 10 different developmental time points, and we will build a web tool for pig skin single-cell RNA dataset by integrating the data of anatomic location and developmental time.

Reviewer #1 (Recommendations for the authors):Some of my suggestions are below and hope they are helpful.1) A huge amount of effort and expense went into creating a dataset from 6 different body sites. It is extremely hard for this reviewer to ascertain from both the writing in the manuscript and the figures what the dataset says about positional identity. Were there differences in the Hox code expression patterns? Did this match what Rinn et al. published on human skin? What were the pathways involved in regulating the phenotypic differences between the tissues? Perhaps a focused study utilizing the data would improve the manuscript.

(1) Hox genes are a family of transcription factor-encoding genes that are crucial for regulating embryonic development and tissue morphogenesis. However, in our study, we did not identify any Hox gene among these differentially expressed genes in skin fibroblasts from both different anatomical sites and different pig populations. The differences of Hox code expression patterns might come from the heterogeneity of different species. Coincidently, some signaling pathways, which related to regional- or populations-specific genes of fibroblasts, were identified in our study, which partially match what Rinn et al. published on human skin. For example, COL11A1 and *MMP3* facilitate ECM accumulation, while DDK2 and GREM are involved in cell signaling guidance, etc.

(2) Moreover, we have further analyzed some valuable data more extensively as follows:

To ascertain the positional identity, in Result Section of “Heterogeneity of skin FBs in different anatomic sites”, we supplemented the gene expression signatures of fibroblasts among diverse anatomic skin sites, and these identified genes were mainly involved in ECM synthesis, cell signaling guidance, metabolism and human diseases pathways. The added results are shown in Figure 5I and Figure 5—figure supplement 2A. Meanwhile, to ascertain the population identity, we have supplemented the single-cell data for SMCs, ECs and FBs between Chenghua and Large White pigs in Result Section of “Heterogeneity of skin cells in different pig populations”, and these added data are shown in Figure 6—figure supplement 1E-G. In addition, the FBs-related gene expression levels of specific-population between the different pig populations have also been added and shown in Figure 6I.

2) The comparison between the Chenghua and Large White is very interesting. The authors note that the thickness of the dermis is different and reveal an analysis that suggests that the expansion of the extra-cellular matrix genes occur in the pig with thicker skin. This is an obvious result. Could the authors expand on the mechanisms of why this would happen? What inherently regulates this difference at a transcriptional level?

We think that the artificial selection and specific feeding environment might contribute to the phenotypic formation of superior skin thickness trait in CH pigs. Moreover, previous study discovered that the dermal collagenous ECM determined individual skin thickness (Qin et al., 2018), so the inherent mechanism might result in the expansion of the extra-cellular matrix genes occur in the CH pigs with thicker skin. In fact, gene expression changes at transcription levels are influenced by multiple regulatory mechanisms, such as transcription factors, epigenetic regulation (including DNA methylation, histone modification, non-coding RNA regulation, etc.), and post-transcriptional regulation of RNA. In our previous studies, we elucidated that the difference of skin thickness between Chenghua and Large White pigs was influenced by the circ004463-miR23b-CADM3/MAP4K4 regulatory axis or miR-218 (Zou et al., 2022, 2023), which belong to non-coding RNA epigenetic regulation. In this study, we found that some transcription factor expansion of the extra-cellular matrix genes occurred in the CH pig with thicker skin and it might contribute to the difference of skin thickness between CH and LW pigs. Some mechanism explanations have been supplemented and added in Result Section of “Heterogeneity of skin cells in different pig populations”.

3) Overall the manuscript could also use editorial help in teasing out the central message of the manuscript. Is this just a resource manuscript allowing for an interesting and important dataset to be shared with the world? If so, the authors should generate a web tool.

In fact, our obtained single-cell data have been uploaded to the GEO database (GEO number: GSE225416) and it is free and available for readers. Although generating a web tool is a nice suggestion, it is difficult for us to achieve it under the present conditions. Because generating a web tool will need much time (about 6 months), money (about 20,000 dollars), and professional technical help. Fortunately, we are now performing single-cell RNA sequencing on CH pig skin during 10 different developmental time points, and we will build a web tool for pig skin single-cell RNA dataset by integrating anatomic location and developmental time data.

4) It would be interesting to create a narrative around the idea that the Chenghua and Large White have a difference in the way that positional identity is regulated between the different species. Is it possible that the authors could show this through their single-cell data?

We have supplemented the single-cell data for SMCs, ECs and FBs between Chenghua and Large White pigs in Result Section of “Heterogeneity of skin cells in different pig populations”, and these added data are shown in Figure 6—figure supplement 1E-G. In addition, the FBs-related gene expression levels of specific-population between the different pig populations have also been added and shown in Figure 6I.

Reviewer #2 (Recommendations for the authors):1. I did not find the age and sex of the pigs. This basic information should be given.

The age and sex of the pigs should be 180-day-old and female, and the information has been added in the Material and Method Section of “Skin samples dissociation and cell collection”.

2. Which skin layer is dissected to separate the specimen from the body? This may lead to different thickness of the skin from different regions. Are all the components dissociated into single cells? Do some cells remain undissociated in the tissue debris? All these may affect the analyses of cell compositions. These differences may be unavoidable but the condition should be clarified. Showing photos of specimens in supplements may help.

In our present study, the full-layer skin was dissected to separate the specimen from the pig body and all the skin components, which thoroughly scrapped off the hair and subcutaneous fat, were dissociated into single cells without the undissociated tissue debris. The related information have been added in the Material and Method Section of “Skin samples dissociation and cell collection”. In addition, the photos of skin specimens have been added and shown in Figure 5—figure supplement 1A and Figure 6—figure supplement 1A.

3. English should be improved. There are also typos. For example, p. 52. Pseudotiom should be pseudotime.

To improve English writing levels and avoid grammatical errors, Professor Jiang (corresponding author) have edited throughout the manuscript. We believe that the language is now acceptable for the review process and understandable to general readers.

3. Panel 2E needs more explanation.

In fact, there are some explanations for Figure 2E. The explanation of Figure 2E might was not clearly written and it puzzled the reviewer's understanding. So, we have edited the related sentences to improve readability.

4. Figure 5A and 6A. H&E staining needs improvement.

We have chosen higher resolution pictures to replace the Figure 5A and 6A H&E staining.

5. A major possibility to improve the significance is to evaluate the functional significance further. It may not be practical for the authors. The current work is still valuable but will have limitations in its impact.

We appreciate the reviewer's suggestions for evaluating the functional significance further. In our next research, we will perform some experiments in vitro and in vivo to explore the functions of these identified key genes.